# RNNs Incrementally Evolving on an Equilibrium Manifold: A Panacea for Vanishing and Exploding Gradients?

**Anil Kag**
ECE Department
Boston University
Boston, MA 02215, USA
anilkag@bu.edu

**Ziming Zhang**
ECE Department
Worcester Polytechnic Institute
Worcester, MA 01609, USA
zzhang15@wpi.edu

**Venkatesh Saligrama**
ECE Department
Boston University
Boston, MA 02215, USA
srv@bu.edu

## Abstract

Recurrent neural networks (RNNs) are particularly well-suited for modeling long-term dependencies in sequential data, but are notoriously hard to train because the error backpropagated in time either vanishes or explodes at an exponential rate. While a number of works attempt to mitigate this effect through gated recurrent units, skip-connections, parametric constraints and design choices, we propose a novel incremental RNN (iRNN), where hidden state vectors keep track of incremental changes, and as such approximate state-vector increments of Rosenblatt's (1962) continuous-time RNNs. iRNN exhibits identity gradients and is able to account for long-term dependencies (LTD). We show that our method is computationally efficient overcoming overheads of many existing methods that attempt to improve RNN training, while suffering no performance degradation. We demonstrate the utility of our approach with extensive experiments and show competitive performance against standard LSTMs on LTD and other non-LTD tasks.

## 1 Introduction

Recurrent neural networks (RNNs) in each round store a hidden state vector, $h_m \in \mathbb{R}^D$, and upon receiving the input vector, $x_{m+1} \in \mathbb{R}^d$, linearly transform the tuple $(h_m, x_{m+1})$ and pass it through a memoryless non-linearity to update the state over $T$ rounds. Subsequently, RNNs output an affine function of the hidden states as its prediction. The model parameters (state/input/prediction parameters) are learnt by minimizing an empirical loss. This seemingly simple update rule has had significant success in learning complex patterns for sequential input data.

Nevertheless, that training RNNs can be challenging, and that performance can be uneven on tasks that require long-term-dependency (LTD), was first noted by Hochreiter (1991), Bengio et al. (1994) and later by other researchers. Pascanu et al. (2013b) attributed this to the fact that the error gradient back-propagated in time (BPTT), for the time-step $m$, is dominated by product of partials of hidden-state vectors, $\prod_{j=m}^{T-1} \frac{\partial h_{j+1}}{\partial h_j}$, and these products typically exhibit exponentially vanishing decay or explosion, resulting in incorrect credit assignment during training and test-time.

**Rosenblatt (1962)**, on whose work we draw inspiration from, introduced continuous-time RNN (CTRNN) to mimic activation propagation in neural circuitry. CTRNN dynamics evolves as follows:

$$\tau \dot{g}(t) = -\alpha g(t) + \phi(U g(t) + W x(t) + b), \, t \geq t_0. \tag{1}$$

Here, $x(t) \in \mathbb{R}^d$ is the input signal, $g(t) \in \mathbb{R}^D$ is the hidden state vector of $D$ neurons, $\dot{g}_i(t)$ is the rate of change of the $i$-th state component; $\tau, \alpha \in \mathbb{R}^+$, referred to as the post-synaptic time-constant, impacts the rate of a neuron's response to the instantaneous activation $\phi(U g(t) + W x(t) + b)$; and $U \in \mathbb{R}^{D \times D}$, $W \in \mathbb{R}^{D \times d}$, $b \in \mathbb{R}^D$ are model parameters. In passing, note that recent RNN works that draw inspiration from ODE's (Chang et al., 2019) are special cases of CTRNN ($\tau = 1$, $\alpha = 0$).

**Vanishing Gradients.** The qualitative aspects of the CTRNN dynamics is transparent in its integral form:

$$g(t) = e^{-\alpha \frac{t-t_0}{\tau}} g(t_0) + \frac{1}{\tau} \int_{t_0}^{t} e^{-\alpha \frac{t-s}{\tau}} \phi(Ug(s) + Wx(s) + b)ds \qquad (2)$$

This integral form reveals that the partials of hidden-state vector with respect to the initial condition, $\frac{\partial g(t)}{\partial g(t_0)}$, gets attenuated rapidly (first term in RHS), and so we face a vanishing gradient problem. We will address this issue later but we note that this is not an artifact of CTRNN but is exhibited by ODEs that have motivated other RNNs (see Sec. 2).

**Shannon-Nyquist Sampling.** A key property of CTRNN is that the time-constant $\tau$ together with the first term $-g(t)$, is in effect a low-pass filter with bandwidth $\alpha\tau^{-1}$ suppressing high frequency components of the activation signal, $\phi((Ug(s)) + (Wx(s)) + b)$. This is good, because, by virtue of the Shannon-Nyquist sampling theorem, we can now *maintain fidelity* of discrete samples with respect to continuous time dynamics, in contrast to conventional ODEs ($\alpha = 0$). Additionally, since high-frequencies are already suppressed, in effect we may assume that the input signal $x(t)$ is slowly varying relative to the post-synaptic time constant $\tau$.

**Equilibrium.** The combination of low pass filtering and slowly time varying input has a significant bearing. The state vector as well as the discrete samples evolve close to the equilibrium state, i.e., $g(t) \approx \phi(Ug(t) + Wx(t) + b)$ under general conditions (Sec. 3).

**Incremental Updates.** Whether or not system is in equilibrium, the integral form in Eq. 2 points to gradient attenuation as a fundamental issue. To overcome this situation, we store and process increments rather than the cumulative values $g(t)$ and propose dynamic evolution in terms of increments. Let us denote hidden state sequence as $h_m \in \mathbb{R}^D$ and input sequence $x_m \in \mathbb{R}^d$. For $m = 1, 2, \ldots, T$, and a suitable $\beta > 0$

$$\tau\dot{g}(t) = -\alpha(g(t) \pm h_{m-1}) + \phi(U(g(t) \pm h_{m-1}) + Wx_m + b), \; g(0) = 0, \; t \geq 0 \quad (3)$$
$$h_m \triangleq h_m^{\beta \cdot \tau} \triangleq g(\beta \cdot \tau)$$

Intuitively, say system is in equilibrium and $-\alpha(\mu(x_m, h_{m-1})) + \phi(U\mu(x_m, h_{m-1}) + Wx_m + b) = 0$. We note state transitions are marginal changes from previous states, namely, $h_m = \mu(x_m, h_{m-1}) - h_{m-1}$. Now for a fixed input $x_m$, as to which equilibrium is reached depends on $h_{m-1}$, but are nevertheless finitely many. So encoding marginal changes as states leads to "identity" gradient.

**Incremental RNN (iRNN) achieves Identity Gradient.** We propose to discretize Eq. 3 to realize iRNN (see Sec. 3). At time $m$, it takes the previous state $h_{m-1} \in \mathbb{R}^D$ and input $x_m \in \mathbb{R}^d$ and outputs $h_m \in \mathbb{R}^D$ after simulating the CTRNN evolution in discrete-time, for a suitable number of discrete steps. We show that the proposed RNN approximates the continuous dynamics and solves the vanishing/exploding gradient issue by ensuring identity gradientIn general, we consider two options, SiRNN, whose state is updated with a single CTRNN sample, similar to vanilla RNNs, and, iRNN, with many intermediate samples. SiRNN is well-suited for slowly varying inputs.

**Contributions.** To summarize, we list our main contributions:
(A) iRNN converges to equilibrium for typical activation functions. The partial gradients of hidden-state vectors for iRNNs converge to identity, thus solving vanishing/exploding gradient problem!
(B) iRNN converges rapidly, at an exponential rate in the number of discrete samplings of Eq. 1. SiRNN, the single-step iRNN, is efficient and can be leveraged for slowly varying input sequences. It exhibits fast training time, has fewer parameters and better accuracy relative to standard LSTMs.
(C) Extensive experiments on LTD datasets show that we improve upon standard LSTM accuracy as well as other recent proposals that are based on designing transition matrices and/or skip connections. iRNNs/SiRNNs are robust to time-series distortions such as noise paddings
(D) While our method extends directly (see Appendix A.1) to Deep RNNs, we deem these extensions complementary, and focus on single-layer to highlight our incremental perspective.

## 2 RELATED WORK

**Gated Architectures.** Long short-term memory (LSTM) (Hochreiter & Schmidhuber, 1997) is widely used in RNNs to model long-term dependency in sequential data. Gated recurrent unit (GRU) (Cho et al., 2014) is another gating mechanism that has been demonstrated to achieve similar

performance of LSTM with fewer parameters. Some recent gated RNNs include UGRNN (Collins et al., 2016), and FastGRNN (Kusupati et al., 2018). While mitigating vanishing/exploding gradients, they do not eliminate it. Often, these models incur increased inference, training costs, and model size.

**Unitary RNNs.** Arjovsky et al. (2016); Jing et al. (2017); Zhang et al. (2018); Mhammedi et al. (2016) focus on designing well-conditioned state transition matrices, attempting to enforce unitary-property, during training. Unitary property does not generally circumvent vanishing gradient (Pennington et al. (2017)). Also, it limits expressive power and prediction accuracy while also increasing training time.

**Deep RNNs.** These are nonlinear transition functions incorporated into RNNs for performance improvement. For instance, Pascanu et al. (2013a) empirically analyzed the problem of how to construct deep RNNs. Zilly et al. (2017) proposed extending the LSTM architecture to allow step-to-step transition depths larger than one. Mujika et al. (2017) proposed incorporating the strengths of both multiscale RNNs and deep transition RNNs to learn complex transition functions. While Deep RNNs offer richer representations relative to single-layers, it is complementary to iRNNs.

**Residual/Skip Connections.** Jaeger et al. (2007); Bengio et al. (2013); Chang et al. (2017); Campos et al. (2017); Kusupati et al. (2018) feed-forward state vectors to induce skip or residual connections, to serve as a middle ground between feed-forward and recurrent models, and to mitigate gradient decay. Nevertheless, these connections, cannot entirely eliminate gradient explosion/decay. For instance, Kusupati et al. (2018) suggest $h_m = \alpha_m h_{m-1} + \beta_m \phi(U h_{m-1} + W x_m + b)$, and learn parameters so that $\alpha_m \approx 1$ and $\beta_m \approx 0$. Evidently, this setting can lead to identity gradient, observe that setting $\beta_m \approx 0$, implies little contribution from the inputs and can conflict with good accuracy, as also observed in our experiments.

**Linear RNNs.** (Bradbury et al., 2016; Lei et al., 2018; Balduzzi & Ghifary, 2016) focus on speeding up RNNs by replacing recurrent connections, such as hidden-to-hidden interactions, with light weight linear components. This reduces training time, but results in significantly increased model size. For example, Lei et al. (2018) requires twice the number of cells for LSTM level performance.

**ODE/Dynamical Perspective.** Few ODE inspired architectures attempt to address stability, but do not end up eliminating vanishing/exploding gradients. Talathi & Vartak (2015) proposed a modified weight initialization strategy based on a dynamical system perspective on weight initialization process to successfully train RNNs composed of ReLUs. Niu et al. (2019) analyzed RNN architectures using numerical methods of ODE and propose a family of ODE-RNNs. Chang et al. (2019), propose Antisymmetric-RNN. Their key idea is to express the transition matrix in Eq. 1, for the special case $\alpha = 0, \tau = 1$, as a difference: $U = V - V^T$ and note that the eigenspectrum is imaginary. Nevertheless, Euler discretization, in this context leads to instability, necessitating damping of the system. As such vanishing gradient cannot be completely eliminated. Its behavior is analogous to FastRNN Kusupati et al. (2018), in that, identity gradient conflicts with high accuracy. In summary, we are the first to propose evolution over the equilibrium manifold, and demonstrating identity gradients. Neural ODEs (Chen et al., 2018; Rubanova et al., 2019) have also been proposed for time-series prediction to deal with irregularly sampled inputs. They parameterize the derivative of the hidden-state in terms of an *autonomous* differential equation and let the ODE evolve in continuous time until the next input arrives. As such, this is not our goal, our ODE explicitly depends on the input, and evolves until equilibrium for that input is reached. We introduce incremental updates to bypass vanishing/exploding gradient issues, which is not of specific concern for these works.

## 3 METHOD

We use Euler's method to discretize Eq. 3 in steps $\delta = \eta \tau$. Denoting the kth step as $g_k = g(k\delta)$

$$\tau \frac{g_k - g_{k-1}}{\delta} = -\alpha(g_{k-1} + h_{m-1}) + \phi(U(g_{k-1} + h_{m-1}) + W x_m + b), \, k \in [K] \quad (4)$$

Rearranging terms we get a compact form for iRNN (see Fig. 1). In addition we introduce a learnable parameter $\eta_m^k$ and let it be a function of time $m$ and the recursion-step $k$.

$$g_k = g_{k-1} + \eta_m^k (\phi(U(g_{k-1} + h_{m-1}) + W x_m + b) - \alpha(g_{k-1} + h_{m-1})), \, k \in [K] \quad (5)$$
$$h_m^K = g_K$$

We run the recursion for $k \in [K]$ with some suitable initial condition. This could be $g_0 = 0$ or initialized to the previous state, i.e., $g_0 = h_{m-1}$ at time $m$.

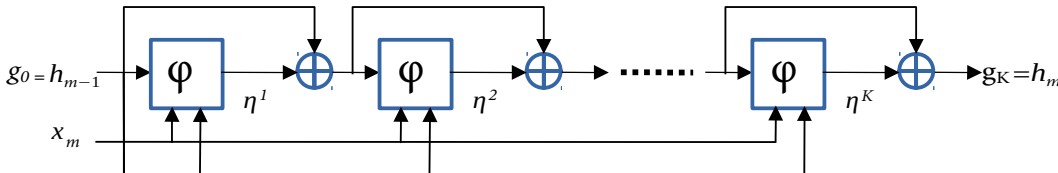

Figure 1: iRNN depicted by unfolding into $K$ recursions for one transition from $g_0 = h_{m-1}$ to $h_m = g_K$. Here, $\varphi(x, g, h) = \phi(U(g + h) + Wx + b) - \alpha(g + h)$. See Sec. A.2 for implementation and pseudo-code. This resembles Graves (2016), who propose to vary $K$ with $m$ as a way to attend to important input transitions. However, the transition functions used are gated units, unlike our conventional ungated functions. As such, while this is not their concern, equilibrium may not even exist and identity gradients are not guaranteed in their setup.

In many of our examples, we find the input sequence is slowly varying, and $K = 1$ can also realize good empirical performance. We refer to this as single-step-incremental-RNN (SiRNN).

$$h_m^1 = g_0 + \eta_m(\phi(U(g_0 + h_{m-1}) + Wx_m + b) - \alpha(h_{m-1} + g_0)) \qquad (6)$$

For both iRNN and SiRNN we drop the superscript whenever it is clear from the context.

**Root Finding and Transitions.** The two indices $k$ and $m$ should not be confused. The index $m \in [T]$ refers to the time index, and indexes input, $x_m$ and hidden state $h_m$ over time horizon $T$. The index $k \in [K]$ is a fixed-point recursion for converging to the equilibrium solution at each time $m$, given input $x_m$ and the hidden state $h_{m-1}$. We iterate over $k$ so that at $k = K$, $g_K$ satisfies,

$$\phi(U(g_K + h_{m-1}) + Wx_m + b) - \alpha(g_K + h_{m-1}) \approx 0$$

The recursion (Eq. 5) at time $m$ runs for $K$ rounds, terminates, and recursion is reset for the new input, $x_{m+1}$. Indeed, Eq. 5 is a standard root-finding recursion, with $g_{k-1}$ serving as the previous solution, plus a correction term, which is the error, $\phi(U(g_{k-1} + h_{m-1}) + Wx_m + b) - \alpha(g_{k-1} + h_{m-1})$. If the sequence converges, the resulting solution is the equilibrium point. Proposition 2 guarantees a geometric rate of convergence.

**Identity Gradient.** We will informally (see Theorem 1) show here that partial gradients are identity. Say we have for sufficiently large $K$, $h_m = g_K$ is the equilibrium solution. It follows that,

$$\phi(U(h_m + h_{m-1}) + Wx_m + b) - \alpha(h_m + h_{m-1})) = 0$$

Taking derivatives, we have,

$$\nabla\phi(\cdot)U\left(\frac{\partial h_m}{\partial h_{m-1}} + I\right) - \alpha\left(\frac{\partial h_m}{\partial h_{m-1}} + I\right) = 0 \implies (\nabla\phi(\cdot)U - \alpha I)\left(\frac{\partial h_m}{\partial h_{m-1}} + I\right) = 0. \quad (7)$$

Thus if the matrix $(\nabla\phi(\cdot)U - \alpha I)$ is not singular, it follows that $(\frac{\partial h_m}{\partial h_{m-1}} + I) = 0$.

**SiRNN vs. iRNN.** SiRNN approximates iRNN. In particular, say $x_m$ is a constant in the segment, $m \in [m_0, m_0 + K]$, then SiRNN trajectory of hidden states, denoted as $h_{m_0+K}^1$ is equal to the iRNN hidden state $h_{m_0}^K$, when both SiRNN and iRNN are initialized with $g_0 = h_{m-1}$. Thus, for slowly time-varying inputs we can expect SiRNN to closely approximate iRNN.

**Residual Connections vs. iRNN/SiRNN.** As such, our architecture is a special case of skip/residual connections. Nevertheless, unlike skip connections, our connections are *structured*, and the dynamics driven by the error term ensures that the hidden state is associated with equilibrium and leads to identity gradient. No such guarantees are possible with unstructured skip connections. Note that for slowly varying inputs, after a certain transition-time period, we should expect SiRNN to be close to equilibrium as well. Without this imposed structure, general residual architectures can learn patterns that can be dramatically different (see Fig. 2).

### 3.1 IDENTITY GRADIENT PROPERTY AND CONVERGENCE GUARANTEES.

Let us now collect a few properties of Eq. 3 and Eq. 5. First, denote the equilibrium solutions for an arbirary input $x \in \mathbb{R}^d$, arbitrary state-vector $\nu \in \mathbb{R}^D$, in an arbitrary round:

$$\mathcal{M}_{eq}(x, \nu) = \{\mu \in \mathbb{R}^D \mid \alpha(\mu + \nu) = \phi(U(\mu + \nu) + Wx + b)\}$$

Whenever the equilibrium set is a singleton, we denote it as a function $h_{eq}(x, \nu)$. For simplicity, we assume below that $\eta_k^i$ is a positive constant independent of $k$ and $i$.

**Proposition 1.** *Suppose, $\phi(\cdot)$ is a 1-Lipshitz function in the norm induced by $\| \cdot \|$, and $\|U\| < \alpha$, then for any $x_m \in \mathbb{R}^d$ and $h_{m-1} \in \mathbb{R}^D$, it follows that $\mathcal{M}_{eq}(x, \nu)$ is a singleton and as $K \to \infty$, the iRNN recursions converge to this solution, namely, $h_m = \lim_{K \to \infty} g_K = h_{eq}(x_m, h_{m-1})$*

*Proof.* Define $T : \mathbb{R}^D \to \mathbb{R}^d$, with $T(g) = (1 - \eta\alpha)g + \eta(\phi(U(g + h_{m-1}) + W x_m + b) - h_{m-1})$. It follows that $T(\cdot)$ is a contraction:

$$\|T(g) - T(g')\| \leq (1 - \eta\alpha)\|g - g'\| + \eta\|\phi(U(g + h_{m-1}) + W x_m + b) - \phi(U(g' + h_{m-1}) + W x_m + b)\|$$
$$\leq (1 - \eta\alpha + \|U\|\eta)\|g - g'\| < \|g - g'\|.$$

We now invoke the Banach fixed point theorem, which asserts that a contractive operator on a complete metric space converges to a unique fixed point, namely, $T^K(g) \to g_*$. Upon substitution, we see that this point $g_*$ must be such that, $\phi(U(g_* + h_{m-1}) + W x_m + b) - (g_* + h_{m-1}) = 0$. Thus equilibrium point exists and is unique. Result follows by setting $h_m \triangleq h_{eq}(x_m, h_{m-1})$. □

**Handling $\|U\| \leq \alpha$.** In experiments, we set $\alpha = 1$, and do not enforce $\|U\| \leq \alpha$ constraint. Instead, we initialize $U$ as a Gaussian matrix with IID mean zero, small variance components. As such, the matrix norm is smaller than 1. Evidently, the resulting learnt $U$ matrix does not violate this condition.

Next we show for $\eta > 0$, iRNN converges at a linear rate, which follows directly from Proposition 1.

**Proposition 2.** *Under the setup in Proposition 1, it follows that,*

$$\|h_m^K - h_{eq}(x_m, h_{m-1})\| \triangleq \|g_K - h_{eq}(x_m, h_{m-1})\| \leq (1 - \alpha\eta + \eta\|U\|)^K \|g_1 - h_{eq}(x_m, h_{m-1})\|$$

**Remark.** Proposition 1 accounts for typical activation functions ReLU, tanh, sigmoids as well as deep RNNs (appendix A.1).

In passing we point out that, in our experiments, we learn parameters $\eta_m^k$, and a result that accounts for this case is desirable. We describe this case in Appendix A.3. A fundamental result we describe below is that partials of hidden-state vectors, on the equilibrium surface is unity. For technical simplicity, we assume a continuously differentiable activation, which appears to exclude ReLU activations. Nevertheless, we can overcome this issue, but requires more technical arguments. The main difficulty stems from ensuring that derivatives along the equilibrium surface exist, and this can be realized by invoking the implicit function theorem (IFT). IFT requires continuous differentiability, which ReLUs violate. Nevertheless, recent results [1] suggests that one can state implicit function theorem for everywhere differentiable functions, which includes ReLUs.

**Theorem 1.** *Suppose $\phi(\cdot)$ is a continuously differentiable, 1-Lipshitz function, with $\|U\| < \alpha$. Then as $K \to \infty$, $\frac{\partial h_m}{\partial h_{m-1}} \to \frac{\partial h_{eq}(x_m, h_{m-1})}{\partial h_{m-1}} = -I$. Furthermore, as $K \to \infty$ the partial gradients over arbitrary number of rounds for iRNN is identity.*

$$\frac{\partial h_r}{\partial h_s} = \prod_{r \geq m > s} \frac{\partial h_m}{\partial h_{m-1}} = (-1)^{r-s}\mathbf{I} \Rightarrow \left\| \frac{\partial h_r}{\partial h_s} \right\| = 1. \tag{8}$$

*Proof.* Define, $\psi(g, h_{m-1}) = \phi(U(g + h_{m-1}) + W x_m + b) - \alpha(g + h_{m-1})$. We overload notation and view the equilibrium point as a function of $h_{m-1}$, i.e., $g_*(h_{m-1}) = h_{eq}(x_m, h_{m-1})$. Invoking standard results [2] in ODE's, it follows that $g_*(h_{m-1})$ is a smooth function, so long as the Jacobian, $\nabla_g \psi(g_*, h_{m-1})$ with respect to the first coordinate, $g_*$, is non-singular. Upon computation, we see that, $\nabla_g \psi(g_*, h_{m-1}) = \nabla\phi(g_*, h_{m-1})U - \alpha I$, is non-singular, since $\|\nabla\phi(g_*, h_{m-1})U\| \leq \|U\|$. It follows that we can take partials of the state-vectors. By taking the partial derivatives *w.r.t.* $h_{m-1}$ in Eq. 5, at the equilibrium points we have $[\nabla\phi(g_*, h_{m-1})U - \alpha\mathbf{I}][\frac{\partial g_*}{\partial h_{m-1}} + \mathbf{I}] = \mathbf{0}$ (see Eq. 7). The rest of the proof follows by observing that the first term is non-singular. □

---

[1] terrytao.wordpress.com/2011/09/12/the-inverse-function-theorem-for-everywhere-differentiable-maps/

[2] http://cosweb1.fau.edu/~jmirelesjames/ODE_course/lectureNotes_shortVersion_day1.pdf

**Remark.** We notice that replacing $h_{m-1}$ with $-h_{m-1}$ in Eq. 12 will lead to $\frac{\partial h_{eq}}{\partial h_{m-1}} = \mathbf{I}$, which also has no impact on magnitudes of gradients. As a result, both choices are suitable for circumventing vanishing or exploding gradients during training, but still may converge to different local minima and thus result in different test-time performance. Furthermore, notice that the norm preserving property is somewhat insensitive to choices of $\alpha$, so long as the non-singular condition is satisfied.

### 3.2 iRNN Design Implications: Low-Rank Model Parametrization

Fig. 2 depicts phase portrait and illustrates salient differences between RNN, FastRNN (RNN with skip connection), and iRNN (K=5). RNN and Fas-

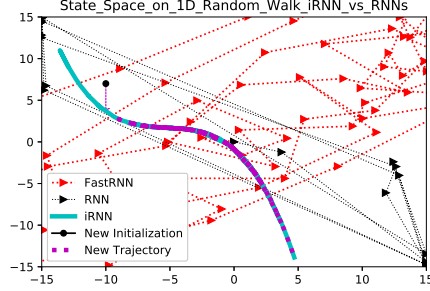

Figure 2: Phase-space trajectory with tanh activation of RNN, FastRNN, iRNN. X-axis denotes 1st dimension, and Y-axis 2nd dimension of 2D hidden state subject to random walk input with variance 10 for 1000 time-steps. Parameters $U, W, b$ are randomly initialized. RNN states are scaled to fit plot since FastRNN is not required to be in the cube.

tRNN exhibit complex trajectories, while iRNN trajectory is smooth, projecting initial point (black circle) onto the equilibrium surface (blue) and moving within it (green). This suggests that iRNN trajectory belongs to a low-dimensional manifold.

**Variation of Equilibrium *w.r.t.* Input.** As before, $h_{eq}$ be an equilibrium solution for some tuple $(h_{m-1}, x_m)$. It follows that,

$$(\alpha \mathbf{I} - \nabla\phi(U(h_{eq} + \mathbf{h}_{m-1}) + Wx_m + b)U)\partial h_{eq} = \nabla\phi(U(h_{eq} + h_{m-1}) + Wx_m + b)W\partial x_m$$

This suggests that, whenever the input undergoes a slow variation, we expect that the equilibrium point moves in such a way that $U\partial h_{eq}$ must lie in a transformed span of $W$. Now $W \in \mathbb{R}^{D \times d}$ with $d \ll D$, which implies that $(\alpha \mathbf{I} - \nabla\phi(U(h_{eq} + h_{m-1}) + Wx_m + b)U$ is rank-deficient.

**Low Rank Matrix Parameterization.** For typical activation functions, note that whenever the argument is in the unsaturated regime, $\nabla\phi(\cdot) \approx \mathbf{I}$. We then approximately get $\mathrm{span}(\alpha\mathbf{I} - U) \approx \mathrm{span}(W)$. We can express these constraints as $U = \alpha\mathbf{I} + VH$ with low-rank matrices $V \in \mathbb{R}^{D \times d_1}, H \in \mathbb{R}^{d_1 \times D}$, and further map both $Uh_m$ and $Wx_m$ onto a shared space. Since in our experiments the signal vectors we encounter are low-dimensional, and sequential inputs vary slowly over time, we enforce this restriction in all our experiments. In particular, we consider,

$$\phi\left(P[U(h_m + h_{m-1}) + Wx_m + b]\right) - (h_m + h_{m-1}) = \mathbf{0}. \tag{9}$$

The parameter matrix $P \in \mathbb{R}^{D \times D}$ maps the contributions from input and hidden states onto the same space. To decrease model-size we let $P = U = (\mathbf{I} + VH)$ learn these parameters.

## 4 Experiments

We organize this section as follows. First, the experimental setup, competing algorithms will be described. Then we present an ablative analysis to highlight salient aspects of iRNN and justify some of our experimental choices. We then plot and tabulate experimental results on benchmark datasets.

### 4.1 Experimental Setup and Baselines

**Choice of Competing Methods:** We choose competing methods based on the following criteria: (a) methods that are devoid of additional application or dataset-specific heuristics, (b) methods that leverage only single cell/block/layer, and (c) methods without the benefit of complementary add-ons (such as gating, advanced regularization, model compression, etc.). Requiring (a) is not controversial since our goal is methodological. Conditions (b),(c) are justifiable since we could also leverage these add-ons and are not germane to any particular method[3]. We benchmark *iRNN* against standard RNN, LSTM (Hochreiter & Schmidhuber, 1997), (ungated) AntisymmetricRNN (Chang et al., 2019), (ungated) FastRNN (Kusupati et al., 2018).

---

[3]These conditions eliminate some potential baselines. We provide specific justifications in the appendix A.5.

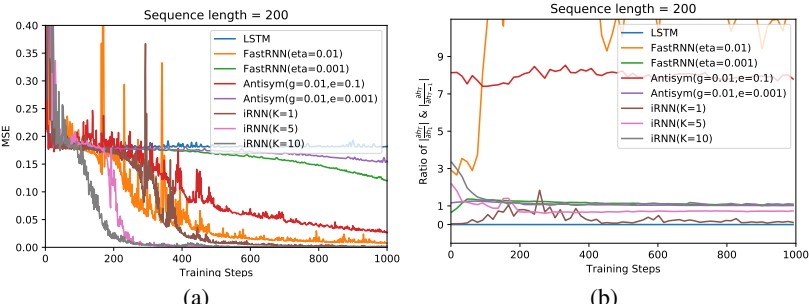

Figure 3: Exploratory experiments for the Add task (a) Convergence with varying K; (b) Ratio $\|\frac{\partial h_T}{\partial h_1}\|/\|\frac{\partial h_T}{\partial h_{T-1}}\|$ illustrates Vanishing/Exploding gradient ($\|\frac{\partial h_T}{\partial h_{T-1}}\|$ and loss gradients are omitted but displayed in A.7.8. For iRNN (a) and (b) together show strong correlation of gradient with accuracy in contrast to other methods.

**Unitary RNN Variants.** Results for methods based on unitary transitions (such as Arjovsky et al. (2016); Wisdom et al. (2016); Vorontsov et al. (2017); Zhang et al. (2018)) are not reported in the main paper (when available reported in appendix) for the following reasons: (a) They are substantially more expensive, and requiring large model sizes; (b) Apart from the benchmark copy and add tasks, results tabulated by FastRNN and Antisymmetric authors (see Zhang et al. (2018); Chang et al. (2019)) show that they are well below SOTA; (c) iRNN dominates unitary-RNN variants on add-task (see Sec. 4.3.1); (d) On copy task, while unitary invariants are superior, Vorontsov et al. (2017) attributes it to modReLU or leaky ReLU activations. Leaky ReLUs allow for linear transitions, and copy task being a memory task benefits from it. With hard non-linear activation, unitary RNN variants can take up to 1000's of epochs for even 100-length sequences (Vorontsov et al. (2017)).

**Implementation.** For all our experiments, we used the parametrized update formulation in Eq. 9 for *iRNN* . We used tensorflow framework for our experiments. For most competing methods apart from AntisymmetricRNN, which we implemented, code is publicly available. All the experiments were run on an Nvidia GTX 1080 GPU with CUDA 9 and cuDNN 7.0 on a machine with Intel Xeon 2.60 GHz CPU with 20 cores.

**Datasets.** Pre-processing and feature extraction details for all publicly available datasets are in the appendix A.4. We replicate benchmark test/train split with 20% of training data for validation to tune hyperparameters. Reported results are based on the full training set, and performance achieved on the publicly available test set. Table 4 (Appendix) and A.4 describes details for all the data sets.

**Hyper Parameters** We used grid search and fine-grained validation wherever possible to set the hyper-parameters of each algorithm, or according to the settings published in (Kusupati et al., 2018; Arjovsky et al., 2016) (*e.g.* number of hidden states). Both the learning rate and $\eta$'s were initialized to $10^{-2}$. The batch size of 128 seems to work well across all the data sets. We used ReLU as the non-linearity and Adam (Kingma & Ba (2015)) as the optimizer for all the experiments.

## 4.2 ABLATIVE ANALYSIS

We perform ablative analysis on the benchmark add-task (Sec 4.3.1) for sequence length 200 for 1000 iterations and explore mean-squared error as a metric. Fig. 3 depicts salient results.

**(a) Identity Gradients & Accuracy:** iRNN accuracy is correlated with identity gradients. Increasing $K$ improves gradients, and correlates with increased accuracy (Fig. 3). While other models $h_t = \alpha h_{t-1} + \beta\phi((U - \gamma I)h_{t-1} + Wx_t)$, can realize identity gradients for suitable choices; linear ($\alpha = 1, \beta = 1, \gamma = 0, U = 0$), FastRNN ($\alpha \approx 1, \beta \approx 0, \gamma = 0$) and Antisymmetric ($\alpha = 1, \beta = 1, U = V - V^T, \|U\| \leq \gamma$), this goal may not be correlated with improved test accuracy. FastRNN($\eta = 0.001$), Antisymmetric ($\gamma = 0.01, \epsilon = 0.001$) have good gradients but poorer test accuracy relative to FastRNN($\eta = 0.01$), Antisymmetric($\gamma = 0.01, \epsilon = 0.1$), with poorer gradients.

**(b) Identity gradient implies faster convergence:** Identity gradient, whenever effective, must be capable of assigning credit to the informative parts, which in turn results in larger loss gradients, and significantly faster convergence with number of iterations. This is borne out in figure 3(a). iRNN for larger $K$ is closer to identity gradient with fewer (unstable) spikes ($K = 1, 5, 10$). With $K = 10$, iRNN converges within 300 iterations while competing methods take about twice this time (other baselines not included here exhibited poorer performance than the once plotted).

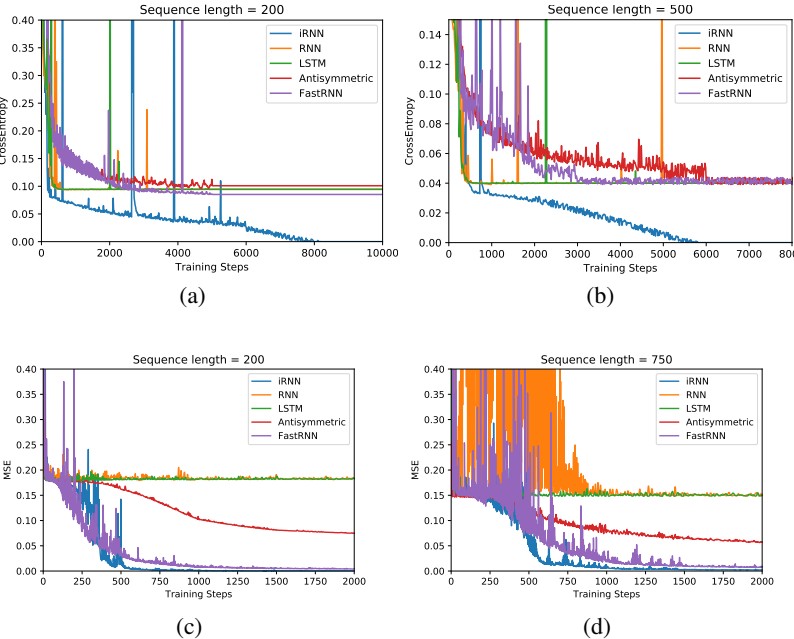

Figure 4: Following Arjovsky et al. (2016) we display average Cross Entropy for the Copy Task (Sequence Length (with baseline memoryless strategy)): (a) 200 (0.09) (b) 500 (0.039). Mean Squared Error for the Add Task, baseline performance is 0.167 (Sequence Length) : (c) 200 (d) 750. For both tasks, iRNN runs $K = 5$.

**(c) SiRNN (iRNN with $K = 1$ delivers good performane in some cases.** Fig. 3(a) illustrates that iRNN $K = \{5, 10\}$ achieves faster convergence than SiRNN, but the computational overhead per iteration roughly doubles or triples in comparison. SiRNN is faster relative to competitors. For this reason, we sometimes tabulate only SiRNN, whenever it is SOTA in benchmark experiments, since accuracy improves with $K$ but requires higher overhead.

### 4.3 LONG-TERM DEPENDENCY AND OTHER TASKS

We list five types of datasets, all of which in some way require effective gradient propagation: (1) Conventional Benchmark LTD tasks (Add & Copy tasks) that illustrate that iRNN can rapidly learn long-term dependence; (2) Benchmark vision tasks (pixel MNIST, perm-MNIST) that may not require long-term, but nevertheless, demonstrates that *iRNN* achieves SOTA for short term dependencies but with less resources. (3) Noise Padded (LTD) Vision tasks (Noisy MNIST, Noisy CIFAR), where a large noise time segment separates information segments and the terminal state, and so the learner must extract information parts while rejecting the noisy parts; (4) short duration activity embedded in a larger time-window (HAR-2, Google-30 in Appendix Table 4 and many others A.7), that usually arise in the context of smart IoT applications and require a small model-size footprint. Chang et al. (2019) further justify (3) and (4) as LTD, because for these datasets where only a smaller unknown segment(s) of a longer sequence is informative. (5) Sequence-sequence prediction tasks (PTB language modeling) that are different from terminal prediction (reported in appendix A.7).

### 4.3.1 STANDARD BENCHMARK LTD TASKS : ADDITION & COPY MEMORY

Addition and Copy tasks (Hochreiter & Schmidhuber, 1997) have long been used as benchmarks in the literature to evaluate LTD (Hori et al., 2017; Zhang et al., 2018; Arjovsky et al., 2016; Martens & Sutskever, 2011). We follow the setup described in Arjovsky et al. (2016) to create the adding and copying tasks. See appendix A.4 for detailed description. For both tasks we run iRNN with $K = 5$.

Figure 4 show the average performance of various methods on these tasks. For the copying task we observe that *iRNN* converges rapidly to the naive baseline and is the only method to achieve zero average cross entropy. For the addition task, both FastRNN and *iRNN* solves the addition task but FastRNN takes twice the number of iterations to reach desired 0 MSE. [4] In both the tasks,

---

[4] Note that LSTM solves the addition problem in Arjovsky et al. (2016) only with more than $10k$ iterations. We only use $2k$ iterations in our experiments to demonstrate the effectiveness of our method.

*iRNN* performance is much more stable across number of online training samples. In contrast, other methods either takes a lot of samples to match *iRNN* 's performance or depict high variance in the evaluation metric. This shows that *iRNN* converges faster than the baselines (to the desired error). *These results demonstrate that* iRNN *easily and quickly learns the long term dependencies* . We omitted reporting unitary RNN variants for Add and Copy task. See Sec. 4.1 for copy task. On Add-task we point out that our performance is superior. In particular, for the longer $T = 750$ length, Arjovsky et al. (2016), points out that MSE does not reach zero, and uRNN is noisy. Others either (Wisdom et al., 2016) do not report add-task or report only for shorter lengths (Zhang et al., 2018).

Table 1: Results for Pixel-by-Pixel MNIST and Permuted MNIST datasets. $K$ denotes pre-defined recursions embedded in graph to reach equilibrium.

| Data set | Algorithm | Accuracy (%) | Train Time (hr) | #Params |
|---|---|---|---|---|
| Pixel-MNIST | FastRNN | 96.44 | 15.10 | 33k |
| | RNN | 94.10 | 45.56 | 14k |
| | LSTM | 97.81 | 26.57 | 53k |
| | Antisymmetric | 98.01 | 8.61 | 14k |
| | *iRNN* (K=1) | 97.73 | **2.83** | **4k** |
| | ***iRNN* (K=3)** | **98.13** | 2.93 | **4k** |
| Permute-MNIST | FastRNN | 92.68 | 9.32 | 8.75k |
| | LSTM | 92.61 | 19.31 | 35k |
| | Antisymmetric | 93.59 | 4.75 | 14k |
| | ***iRNN* (K=1)** | **95.62** | **2.41** | **8k** |

### 4.3.2 Non LTD Vision Tasks: Pixel MNIST, Permute MNIST

Next, we perform experiments on the sequential vision tasks: (a) classification of MNIST images on a pixel-by-pixel sequence; (b) a fixed random permuted MNIST sequence (Lecun et al., 1998). These tasks typically do not fall in the LTD categories (Chang et al., 2019), but are useful to demonstrate faster training, which can be attributed to better gradients.

For the pixel-MNIST task, Kusupati et al. (2018) reports that it takes significantly longer time for existing (LSTMs, Unitary, Gated, Spectral) RNNs to converge to reasonable performance. In contrast, FastRNN trains at least $2x$ faster than LSTMs. Our results (table 1) for *iRNN* shows a $9x$ speedup relative LSTMs, and $2x$ speedup in comparison to Antisymmetric. In terms of test accuracy, *iRNN* matches the performance of Antisymmetric, but with at least $3x$ fewer parameters. We did not gain much with increased K values[5]. For the permuted version of this task, we seem to outperform the existing baselines [6]. In both tasks, *iRNN* trained at least $2x$ faster than the strongest baselines. *These results demonstrate that* iRNN *converges much faster than the baselines with fewer parameters.*

### 4.3.3 Noise padding Tasks: Noisy-MNIST, Noisy-CIFAR

Additionally, as in Chang et al. (2019), we *induce* LTD by padding CIFAR-10 with noise exactly replicating their setup, resulting in Noisy-CIFAR. We extend this setting to MNIST dataset resulting in Noisy-MNIST. Intuitively we expect our model to be resilient to such perturbations. We attribute iRNN's superior performance to the fact that it is capable of suppressing noise. For example, say noise is padded at $t > \tau$ and this results in $W x_t$ being zero on average. For *iRNN* the resulting states ceases to be updated. So *iRNN* recalls last informative state $h_\tau$ (modulo const) unlike RNNs/variants! Thus information from signal component is possibly better preserved.

Results for Noisy-MNIST and Noisy-CIFAR are shown in Table 2. Note that almost all timesteps contain noise in these datasets. LSTMs perform poorly on these tasks due to vanishing gradients. This

---

[5] For some existing comparisons LSTM have achieved roughly 98.9 with dataset specific heuristics (Cooijmans et al., 2016), but we could not achieve this performance in our comparison (and so have many others like (Kusupati et al., 2018; Zhang et al., 2018; Arjovsky et al., 2016)).

[6] Note that there's no standard permutation in the literature. This may be the main reason we could not replicate Chang et al. (2019) performance on the permute MNIST task.

Table 2: Results for Noise Padded CIFAR-10 and MNIST datasets. Since the equilibrium surface is smooth and resilient to small perturbations, *iRNN* achieves better performance than the baselines with faster convergence.

| Data set | Algorithm | Accuracy (%) | Train Time (hr) | #Params |
|---|---|---|---|---|
| Noisy-MNIST | FastRNN | 98.12 | 8.93 | 11k |
| | LSTM | 10.31 | 19.43 | 44k |
| | Antisymmetric | 97.76 | 5.21 | 10k |
| | *iRNN* (K=1) | **98.48** | **2.39** | **6k** |
| Noisy-CIFAR | FastRNN | 45.76 | 11.61 | 16k |
| | LSTM | 11.60 | 23.47 | 64k |
| | Antisymmetric | 48.63 | 5.81 | 16k |
| | *iRNN* (K=1) | **54.50** | **2.47** | **11.5k** |

is consistent with the earlier observations (Chang et al., 2019). *iRNN* outperforms the baselines very comprehensively on CIFAR-10, while on MNIST the gains are smaller, as it's a relatively easier task. *These results show that* iRNN *is more resilient to noise and can account for longer dependencies.*

Table 3: Results for Activity Recognition Datasets. *iRNN* outperforms the baselines on all metrics even with $K = 1$. Its worth noticing that although $K = 5$ increases test time, it's well within LSTM's numbers, the overall train time and resulting performance are better than $K = 1$.

| Data set | Algorithm | Accuracy (%) | Train Time (hr) | #Params | Test Time (ms) |
|---|---|---|---|---|---|
| HAR-2 | FastRNN | 94.50 | 0.063 | 7.5k | **0.01** |
| | RNN | 91.31 | 0.114 | 7.5k | **0.01** |
| | LSTM | 93.65 | 0.183 | 16k | 0.04 |
| | Antisymmetric | 93.15 | 0.087 | 7.5k | **0.01** |
| | *iRNN* (K=1) | 95.32 | 0.061 | **4k** | **0.01** |
| | *iRNN* (K=5) | **96.30** | **0.018** | **4k** | 0.03 |
| Google-30 | FastRNN | 91.60 | 1.30 | 18k | **0.01** |
| | RNN | 80.05 | 2.13 | 12k | **0.01** |
| | LSTM | 90.31 | 2.63 | 41k | 0.05 |
| | Antisymmetric | 90.91 | 0.54 | 12k | **0.01** |
| | *iRNN* (K=1) | 93.77 | **0.44** | **8.5k** | **0.01** |
| | *iRNN* (K=5) | **94.23** | **0.44** | **8.5k** | 0.05 |

### 4.3.4 SHORT DURATION EMBEDDED ACTIVITY RECOGNITION TASKS: HAR-2, GOOGLE-30

We are interested in detecting activity embedded in a longer sequence with small footprint RNNs (Kusupati et al. (2018)): (a) Google-30 (Warden, 2018), *i.e.* detection of utterances of 30 commands plus background noise and silence, and (b) HAR-2 (Anguita et al., 2012), *i.e.* Human Activity Recognition from an accelerometer and gyroscope on a Samsung Galaxy S3 smartphone.

Table 3 shows accuracy, training time, number of parameters and prediction time. Even with $K = 1$, we compare well against competing methods, and iRNN accuracy improves with larger $K$. Interestingly, higher $K$ yields faster training as well as moderate prediction time, despite the overhead of additional recursions. *These results show that* iRNN *outperforms baselines on activity recognition tasks, and fits within IoT/edge-device budgets.*

## 5 CONCLUSION

Drawing inspiration from Rosenblatts Continuous RNNs, we developed discrete time incremental RNN (iRNN). Leveraging equilibrium properties of CTRNN, iRNN solves exploding/vanishing gradient problem. We show that iRNN improved gradients are directly correlated with improved test accuracy. A number of experiments demonstrate iRNNs responsiveness to long-term dependency tasks. In addition, due to its smooth low-dimensional trajectories, it has a lightweight footprint that can be leveraged for IoT applications.

ACKNOWLEDGMENTS

The authors would like to thank the Area Chair and the reviewers for their constructive comments. This work was supported partly by the National Science Foundation Grant 1527618, the Office of Naval Research Grant N0014-18-1-2257 and by a gift from ARM corporation.

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

## A  APPENDIX

### A.1  MULTI-LAYER DEEP RNN NETWORKS.

We point out in passing that our framework readily admits deep multi-layered networks within a single time-step. Indeed our setup is general; it applies to shallow and deep nets; small and large time steps. As a case in point, the Deep Transition RNN Pascanu et al. (2013c):

$$h_{m+1} = f_h(h_m, x_{m+1}) = \phi_h(W_L \phi_{L-1}(W_{L-1} \dots W_1 \phi_1(U h_m + W x_{m+1})))$$

is readily accounted by Theorem 1 in an implicit form:

$$h_{m+1} = f_h(h_{m+1} + h_m, x_{m+1}) - h_m.$$

So is Deep-RNN Hermans & Schrauwen (2013). The trick is to transform $h_m \to h_m + h_{m+1}$ and $h_{m+1} \to h_m + h_{m+1}$. As such, all we need is smoothness of $f_h$, which has no restriction on # layers. On the other hand, that we do not have to limit the number of time steps is the *point* of Theorem 1, which asserts that the partial differential of hidden states (which is primarily why vanishing/exploding gradient arises Pascanu et al. (2013b) in the first place) is identity!!

### A.2  PSEUDO CODE AND IMPLEMENTATION

Given an input sequence and iRNN model parameters, the hidden states can be generated with the help of subroutine 1. This routine can be plugged into standard deep learning frameworks such as Tensorflow/PyTorch to learn the model parameters via back-propagation.

---

**Algorithm 1:** Pseudo Code for computing iRNN hidden states for one input sequence

---

**Data:** Input sequence $\{x_m\}_{m=1}^T$
**Require:** Number of recursion steps $K$, Model parameters $\left(U, W, b, \alpha, \{\eta_m^k\}\right)$
1 Initial hidden state $h_0 = 0$
2 **for** $m = 1$ **to** $T$ **do**
3   Initialize $g_0$ to zero or $h_{m-1}$
4   **for** $k = 1$ **to** $K$ **do**
5     $g_k = g_{k-1} + \eta_m^k \big(\phi(U(g_{k-1} + h_{m-1}) + W x_m + b) - \alpha(g_{k-1} + h_{m-1})\big)$
6   $h_m = g_K$
**Output:** hidden states $\{h_m\}_{m=1}^T$

---

Table 4: Dataset Statistics & Long Term Dependence

| Dataset | Avg. Activity Time | Input Time | Sequence Ratio | #Train | #Fts | #Steps | #Test |
|---|---|---|---|---|---|---|---|
| Google-30 | 25ms | 1000ms | 3/99 | 51,088 | 32 | 99 | 6,835 |
| HAR-2 | 256ms | 2560ms | 13/128 | 7,352 | 9 | 128 | 2,947 |
| Noisy-MNIST | 28 | 1000 | 7/250 | 60,000 | 28 | 1000 | 10,000 |
| Noisy-CIFAR | 32 | 1000 | 4/125 | 60,000 | 96 | 1000 | 10,000 |
| Pixel-MNIST | | | | 60,000 | 1 | 784 | 10,000 |
| Permuted-MNIST | | | | 60,000 | 1 | 784 | 10,000 |

## A.3 CONVERGENCE GUARANTEES FOR GENERAL LEARNING RATES.

**Theorem 2** (Local Convergence with Linear Rate). *Assume that the function $F(g_i) \triangleq \phi(U(g_i + h_{k-1}) + W x_k + b) - (g_i + h_{k-1})$ and the parameter $\eta_k^{(i)}$ in Eq. 5 satisfies*

$$[\eta_k^{(i)}]^2 \|\nabla F(g_i) F(g_i)\|^2 + 2\eta_k^{(i)} F(g_i)^\top \nabla F(g_i) F(g_i) < 0, \forall k, \forall i. \tag{10}$$

*Then there exists $\epsilon > 0$ such that if $\|g_0 - h_{eq}\| \le \epsilon$ where $h_{eq}$ denotes the fixed point, the sequence $g_i$ generated by the Euler method converges to the equilibrium solution in $\mathcal{M}_{eq}(h_{k-1}, x_k)$ locally with linear rate.*

The proof is based on drawing a connection between the Euler method and inexact Newton methods, and leverages Thm. 2.3 in Dembo et al. (1982). See appendix Sec. A.8.1 Thm. 3 and Sec. A.7.5 (for proof, empirical verification).

**Corollary 1.** *If $\|\mathbf{I} + \eta_k^{(i)} \nabla F(g_i)\| < 1, \forall k, \forall i$, the forward propagation (Eq. 13) is stable and the sequence $\{g_i\}$ converges locally at a linear rate.*

The proof is based on Thm. 2.3 in Dembo et al. (1982), Thm. 2 and Prop. 2 in Chang et al. (2019). See appendix A.8.1 Corollary. 2

## A.4 DATASET DETAILS

Table 4 and table 6 lists the statistics of all the datasets described below.

**Google-12 & Google-30**: Google Speech Commands dataset contains 1 second long utterances of 30 short words (30 classes) sampled at 16KHz. Standard log Mel-filter-bank featurization with 32 filters over a window size of 25ms and stride of 10ms gave 99 timesteps of 32 filter responses for a 1-second audio clip. For the 12 class version, 10 classes used in Kaggle's Tensorflow Speech Recognition challenge[7] were used and remaining two classes were noise and background sounds (taken randomly from remaining 20 short word utterances). Both the datasets were zero mean - unit variance normalized during training and prediction.

---

[7]https://www.kaggle.com/c/tensorflow-speech- recognition-challenge

**HAR-2**[8]: Human Activity Recognition (HAR) dataset was collected from an accelerometer and gyroscope on a Samsung Galaxy S3 smartphone. The features available on the repository were directly used for experiments. The 6 activities were merged to get the binarized version. The classes Sitting, Laying, Walking_Upstairs and Standing, Walking, Walking_Downstairs were merged to obtain the two classes. The dataset was zero mean - unit variance normalized during training and prediction.

**Penn Treebank**: 300 length word sequences were used for word level language modeling task using Penn Treebank (PTB) corpus. The vocabulary consisted of 10,000 words and the size of trainable word embeddings was kept the same as the number of hidden units of architecture. This is the setup used in (Kusupati et al., 2018; Zhang et al., 2018).

**Pixel-MNIST**: Pixel-by-pixel version of the standard MNIST-10 dataset [9]. The dataset was zero mean - unit variance normalized during training and prediction.

**Permuted-MNIST**: This is similar to Pixel-MNIST, except its made harder by shuffling the pixels with a fixed permutation. We keep the random seed as 42 to generate the permutation of 784 pixels.

**Noisy-MNIST**: To introduce more long-range dependencies to the Pixel-MNIST task, we define a more challenging task called the Noisy-MNIST, inspired by the noise padded experiments in Chang et al. (2019). Instead of feeding in one pixel at one time, we input each row of a MNIST image at every time step. After the first 28 time steps, we input independent standard Gaussian noise for the remaining time steps. Since a MNIST image is of size 28 with 1 RGB channels, the input dimension is m = 28. The total number of time steps is set to T = 1000. In other words, only the first 28 time steps of input contain salient information, all remaining 972 time steps are merely random noise. For a model to correctly classify an input image, it has to remember the information from a long time ago. This task is conceptually more difficult than the pixel-by-pixel MNIST, although the total amount of signal in the input sequence is the same.

**Noisy-CIFAR**: This is exactly replica of the noise paded CIFAR task mentioned in Chang et al. (2019). Instead of feeding in one pixel at one time, we input each row of a CIFAR-10 image at every time step. After the first 32 time steps, we input independent standard Gaussian noise for the remaining time steps. Since a CIFAR-10 image is of size 32 with three RGB channels, the input dimension is m = 96. The total number of time steps is set to T = 1000. In other words, only the first 32 time steps of input contain salient information, all remaining 968 time steps are merely random noise. For a model to correctly classify an input image, it has to remember the information from a long time ago. This task is conceptually more difficult than the pixel-by-pixel CIFAR-10, although the total amount of signal in the input sequence is the same.

**Addition Task**: We closely follow the adding problem defined in (Arjovsky et al., 2016; Hochreiter & Schmidhuber, 1997) to explain the task at hand. Each input consists of two sequences of length T. The first sequence, which we denote $x$, consists of numbers sampled uniformly at random $\mathcal{U}[0, 1]$. The second sequence is an indicator sequence consisting of exactly two entries of 1 and remaining entries 0. The first 1 entry is located uniformly at random in the first half of the sequence, whilst the second 1 entry is located uniformly at random in the second half. The output is the sum of the two entries of the first sequence, corresponding to where the 1 entries are located in the second sequence. A naive strategy of predicting 1 as the output regardless of the input sequence gives an expected mean squared error of 0.167, the variance of the sum of two independent uniform distributions.

**Copying Task**: Following a similar setup to (Arjovsky et al., 2016; Hochreiter & Schmidhuber, 1997), we outline the copy memory task. Consider 10 categories, $\{a_i\}_{i=0}^{9}$. The input takes the form of a $T + 20$ length vector of categories, where we test over a range of values of T. The first 10 entries are sampled uniformly, independently and with replacement from $\{a_i\}_{i=0}^{7}$, and represent the sequence which will need to be remembered. The next $T - 1$ entries are set to $a_8$, which can be thought of as the 'blank' category. The next single entry is $a_9$, which represents a delimiter, which should indicate to the algorithm that it is now required to reproduce the initial 10 categories in the output. The remaining 10 entries are set to $a_8$. The required output sequence consists of $T + 10$ repeated entries of $a_8$, followed by the first 10 categories of the input sequence in exactly the same order. The goal is to minimize the average cross entropy of category predictions at each time step of

---

[8]https://archive.ics.uci.edu/ml/datasets/human+activity+recognition+using+ smartphones

[9]http://yann.lecun. com/exdb/mnist/

the sequence. The task amounts to having to remember a categorical sequence of length 10, for T time steps.

A simple baseline can be established by considering an optimal strategy when no memory is available, which we deem the memoryless strategy. The memoryless strategy would be to predict $a_8$ for $T + 10$ entries and then predict each of the final 10 categories from the set $\{a_i\}_{i=0}^{7}$ i=0 independently and uniformly at random. The categorical cross entropy of this strategy is $\frac{10 \log(8)}{T+20}$

**DSA-19**[10]: This dataset is based on Daily and Sports Activity (DSA) detection from a resource-constrained IoT wearable device with 5 Xsens MTx sensors having accelerometers, gyroscopes and magnetometers on the torso and four limbs. The features available on the repository were used for experiments. The dataset was zero mean - unit variance normalized during training and prediction.

**Yelp-5**: Sentiment Classification dataset based on the text reviews[11]. The data consists of 500,000 train points and 500,000 test points from the first 1 million reviews. Each review was clipped or padded to be 300 words long. The vocabulary consisted of 20000 words and 128 dimensional word embeddings were jointly trained with the network.

## A.5 BASELINE JUSTIFICATION

In our experiments section, we stated that some of the potential baselines were removed due to experimental conditions enforced in the setup. Here we clearly justify our choice. Mostly the reasoning is to avoid comparing complementary add-ons and compare the bare-bone cells.

- Cooijmans et al. (2016) is removed since its an add-on and can be applied to any method. Besides its pixel-mnist results involve dataset specific heuristics.

- Gong et al. (2018) is also an add-on and hence can be applied to any method.

- Zilly et al. (2017); Pascanu et al. (2013a); Mujika et al. (2017) denote deep transitioning methods. They are add-ons for any single recurrent block and hence can be applied to any recurrent cell.

- Gating variants of single recurrent cells (Chang et al., 2019; Kusupati et al., 2018) have also been removed. Since *iRNN* can be extended to a gating variant and hence its just an add-on.

Table 5: Various hyper-parameters to reproduce results

| Dataset | Hidden Units |
|---------|--------------|
| Google-30 | 80 |
| HAR-2 | 80 |
| Pixel-MNIST | 128 |
| Permuted-MNIST | 128 |
| Noisy-MNIST | 128 |
| Noisy-CIFAR | 128 |
| Addition Task | 128 |
| Copying Task | 128 |
| PTB | 256 |

## A.6 HYPER-PARAMETERS FOR REPRODUCIBILITY

We report various hyper-parameters we use in our experiments for reproduciblity. As mentioned earlier we mainly use 'ReLU' as the non-linearity and Adam as the optimizer. Apart from this, other hyper-parameters are mentioned in table 5.

Table 6: Other Dataset Statistics & Long Term Dependence

| Dataset | Avg. Acitivity Time | Input Time | Sequence Ratio | #Train | #Fts | #Steps | #Test |
|---------|------------|-----------|----------------|--------|------|--------|-------|
| Google-12 | 25ms | 1000ms | 3/99 | 22,246 | 32 | 99 | 3,081 |
| DSA-19 | 500ms | 5000ms | 13/125 | 4,560 | 45 | 125 | 4,560 |
| Yelp-5 | 20 | 300 | 1/15 | 500,000 | 128 | 300 | 500,000 |
| PTB | | | | 929,589 | 300 | 300 | 82,430 |

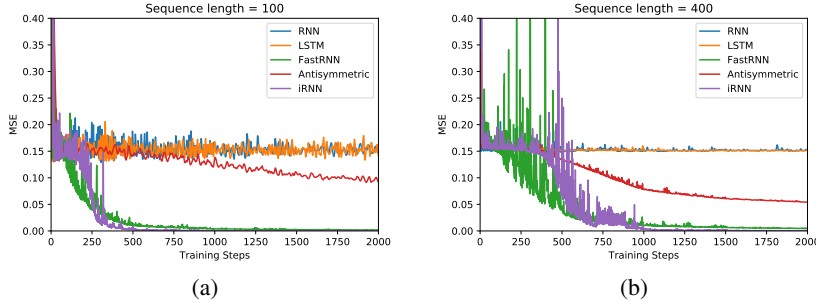

Figure 5: Mean Squared Error shown for the Add Task (Sequence Length) : (c) 100 (d) 400

## A.7 ADDITIONAL EXPERIMENTS

### A.7.1 COPYING AND ADDITION TASKS

Figure 5 shows the results for remaining experiments for the addition task for length $100, 400$.

Table 7: Results for Pixel-by-Pixel MNIST and Permuted MNIST datasets. $K$ denotes pre-defined recursions embedded in graph to reach equillibrium.

| Data set | Algorithm | Accuracy (%) | Train Time (hr) | #Params |
|----------|-----------|--------------|-----------------|---------|
| Pixel-MNIST | FastRNN | 96.44 | 15.10 | 33k |
| | FastGRNN-LSQ | **98.72** | 12.57 | 14k |
| | RNN | 94.10 | 45.56 | 14k |
| | SpectralRNN | 97.7 | | 6k |
| | LSTM | 97.81 | 26.57 | 53k |
| | URNN | 95.1 | | 16k |
| | Antisymmetric | 98.01 | 8.61 | 14k |
| | *iRNN* (K=1) | 97.73 | **2.83** | **4k** |
| | *iRNN* (K=2) | 98.13 | 3.11 | **4k** |
| | ***iRNN* (K=3)** | 98.13 | 2.93 | **4k** |
| Permute-MNIST | FastRNN | 92.68 | 9.32 | 8.75k |
| | SpectralRNN | 92.7 | | 8.5k |
| | LSTM | 92.61 | 19.31 | 35k |
| | URNN | 91.4 | | 12k |
| | Antisymmetric | 93.59 | 4.75 | 14k |
| | ***iRNN* (K=1)** | **95.62** | **2.41** | **8k** |

---

[10]https://archive.ics.uci.edu/ml/datasets/Daily+and+Sports+Activities
[11]https://www.yelp.com/dataset/challenge

### A.7.2 Traditional Datasets

Table 7 shows the results including left out baselines for Pixel-MNIST and permute-MNIST task. Here we also include star rating prediction on a scale of 1 to 5 of Yelp reviews Yelp (2017). Table 8 shows the results for this dataset.

Table 8: Results for Yelp Dataset.

| Data set | Algorithm | Accuracy (%) | Model Size (KB) | Train Time (hr) | Test Time (ms) | #Params |
|---|---|---|---|---|---|---|
| Yelp-5 | FastRNN | 55.38 | 130 | 3.61 | **0.4** | 32.5k |
| | FastGRNN-LSQ | **59.51** | 130 | 3.91 | 0.7 | 32.5k |
| | FastGRNN | 59.43 | 8 | 4.62 | | |
| | RNN | 47.59 | 130 | 3.33 | **0.4** | 32.5k |
| | SpectralRNN | 56.56 | **89** | 4.92 | 0.3 | **22k** |
| | EURNN | 59.01 | 122 | 72.00 | | |
| | LSTM | 59.49 | 516 | 8.61 | 1.2 | 129k |
| | GRU | 59.02 | 388 | 8.12 | 0.8 | 97k |
| | Antisymmetric | 54.14 | 130 | 2.61 | **0.4** | 32.5k |
| | UGRNN | 58.67 | 258 | 4.34 | | |
| | *iRNN* (K=1) | 58.16 | 97.67 | **0.31** | 0.4 | 25k |
| | *iRNN* (K=2) | 59.01 | 98.84 | **0.31** | 0.7 | 25k |
| | *iRNN* (K=3) | 59.34 | 100 | 1.16 | 1.0 | 25k |

### A.7.3 Activity Recognition Datasets

We also include activity recognition tasks: (a)Google-12 Warden (2018) , *i.e.* detection of utterances of 10 commands plus background noise and silence and (b) DSA-19 Altun et al. (2010), Daily and Sports Activity (DSA) detection from a resource-constrained IoT wearable device with 5 Xsens MTx sensors having accelerometers, gyroscopes and magnetometers on the torso and four limbs. Table 9 shows results for these activities along with some other baselines for activity recognition tasks mentioned in Sec. 4.3.4 and described in Sec. A.4.

### A.7.4 PTB Language Modelling

We follow (Kusupati et al., 2018; Zhang et al., 2018) to setup our PTB experiments. We only pursue one layer language modelling, but with more difficult sequence length (300). Table 10 reports all the evaluation metrics for the PTB Language modelling task with 1 layer as setup by Kusupati et al. (2018), including test time and number of parameters (which we omitted from the main paper due to lack of space).

### A.7.5 Linear Rate of Convergence to Fixed Poi

Empirically we verify the local convergence to a fixed point with linear rate by comparing the Euclidean distance between the approximate solutions, $\mathbf{h}_t^{(k)}$, using Eq. 11 with $g_0 = 0$ and the fixed points, $\mathbf{h}_t$, computed using FSOLVE from SCIPY. The learnable parameters are initialized suitably and then fixed. We illustrate our results in Fig. 6, which clearly demonstrates that the approximate solutions tend to converge with linear rate.

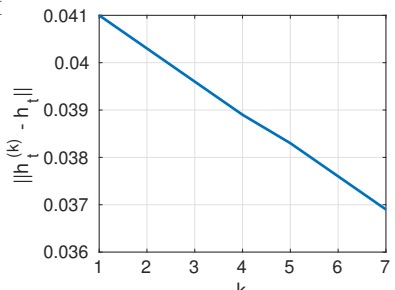

Figure 6: Linear convergence in *iRNN* .

$$g_i = g_{i-1} + \eta_t^i(\phi(U(g_{i-1} + h_{t-1}) + W x_t + b) - \alpha(g_{i-1} + h_{t-1})) \qquad (11)$$
$$h_t^K = g_K$$

### A.7.6 THEORETICAL VERIFICATION

Here we include some experiments to show that our theoretical assumptions hold true.

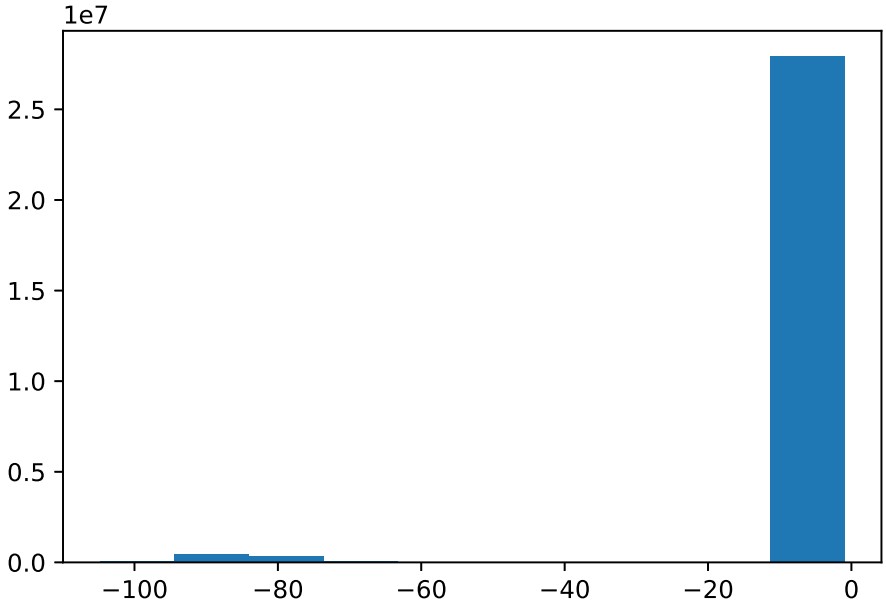

Figure 7: Histogram of the eigenvalues of $\nabla\phi\mathbf{U} - \mathbf{I}$ for *iRNN* on HAR-2 dataset.

**Non-Singularity of the matrix D** For our *iRNN* parametrization to satisfy the conditions of having equillibrium points to be locally asymptotically stable, the eigen values of the matrix $D = (\nabla\phi(\cdot)U - \gamma I)$ should be negative. We plot a histogram of the eigenvalues of $D$ for all the points in the HAR-2 dataset. As illustrated in the figure 7, all the eigenvalues are negative.

### A.7.7 IDENTITY GRADIENT COMPARISON *iRNN* VS RNN

To verify Theorem. 1 empirically, we train RNN and iRNN on the HAR-2 data set (see more details in Sec. 4), respectively, and plot in Fig. 8 the magnitude of gradient of the last layer $\mathbf{h}_T$ *w.r.t.* the first

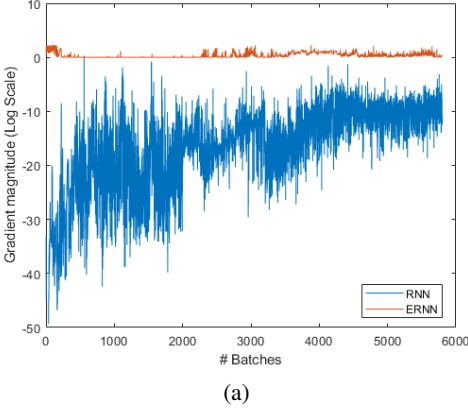

(a)

Figure 8: Comparison between RNN and iRNN on the magnitudes of gradients.

layer $\mathbf{h}_1$ in log scale to confirm that our approach leads to no vanishing or exploding gradients when the error is back-propagated through time. We also conducted experiments to verify that the gradient of iRNN is norm preserving (see Sec. A.7.8 and Figure . 3). As we see clearly, RNN suffers from serious vanishing gradient issue in training, while iRNN's backpropagated gradients is close to 1, and the variance arises mainly our approximation of fixed points and stochastic behavior in training networks, demonstrating much better training stability of iRNN.

### A.7.8   GRADIENT NORM W.R.T. LOSS $\|\frac{\partial L}{\partial h_1}\|$

In addition to the gradient ratio we plot in Sec.4.2, we also show in figure 9, the more popular quantity captured in earlier works (Arjovsky et al., 2016; Zhang et al., 2018), i.e. the gradient norm w.r.t. loss $\|\frac{\partial L}{\partial h_1}\|$. We emphasize that this quantity alone is misleading in the context of resolving the issue of vanishing/exploding gradients. Since $\|\frac{\partial L}{\partial h_1}\| = \|\frac{\partial L}{\partial h_T}\| * \|\frac{\partial h_T}{\partial h_1}\|$. The long term component controlling the gradients is $\|\frac{\partial h_T}{\partial h_1}\|$, but the other component, $\|\frac{\partial L}{\partial h_T}\|$ could become zero by the virtue that the loss is nearly zero. This happens in our addition task experiment, because MSE is close to zero, we experience nearly 0 value for this quantity. But this is clearly because the MSE is 0. Also note that none of our graphs have log scale, which is not the case in earlier works. The conclusion that can be drawn from the loss-gradient is that it is somewhat stable, and can inform us about quality of convergence.

We also plot $\|\frac{\partial h_T}{\partial h_{T-1}}\|$ in figure 9 in order to show that indeed *iRNN* achieves identity gradients everywhere in the time horizon, since fig. 3 had shown that the ratio of $\|\frac{\partial h_T}{\partial h_1}\|$ and $\|\frac{\partial h_T}{\partial h_{T-1}}\|$ equals 1 for *iRNN* .

### A.7.9   DIFFERENT ACTIVATION FUNCTION

We also performed some experiments for sigmoid activation on HAR-2 dataset. The results for this variant also follow similar pattern as we saw in ReLU variant.

### A.8   PROOFS

### A.8.1   LOCAL CONVERGENCE WITH LINEAR RATE

Recall that we rewrite the fixed-point constraints in our *iRNN* as the following ODE:

$$g'_k(t) = F(g_i) \stackrel{def}{=} \phi(U(g_i + h_{k-1}) + W x_k + b) - (g_i + h_{k-1}); \ g(0) = 0. \tag{12}$$

Then based on the Euler method, we have the following update rule for solving fixed-points:

$$g_{i+1} = g_i + \eta_k^{(i)} F(g_i) \tag{13}$$

$$= g_i + \eta_k^{(i)} [\phi(U(g_i + h_{k-1}) + W x_k + b) - (g_i + h_{k-1})]. \tag{14}$$

*Inexact Newton methods* Dembo et al. (1982) refer to a family of algorithms that aim to solve the equation system $F(z) = 0$ approximately at each iteration using the following rule:

$$z_{i+1} = z_i + s_i, \ r_i = F(z_i) + \nabla F(z_i) s_i, \tag{15}$$

where $\nabla F$ denotes the (sub)gradient of function $F$, and $r_i$ denotes the error at the $i$-th iteration between $F(z_i)$ and 0.

By drawing the connection between Eq. 13 and Eq. 15, we can set $z_i \equiv g_i$ and $s_i \equiv \eta_k^{(i)} F(g_i)$. Then based on Eq. 15 we have

$$r_i = F(g_i) + \eta_k^{(i)} \nabla F(g_i) F(g_i). \tag{16}$$

**Lemma 1** (Thm. 2.3 in Dembo et al. (1982)). *Assume that*

$$\frac{\|r_i\|}{\|F(z_i)\|} \leq \tau < 1, \forall k, \tag{17}$$

*where $\| \cdot \|$ denotes an arbitrary norm and the induced operator norm. There exists $\varepsilon > 0$ such that, if $\|z_0 - z_*\| \leq \varepsilon$, then the sequence of inexact Newton iterates $\{z_i\}$ converges to $z_*$. Moreover, the convergence is linear in the sense that $\|z_{i+1} - z_*\|_* \leq \tau \|z_i - z_*\|_*$, where $\|y\|_* = \|\nabla F(z_*)y\|$.*

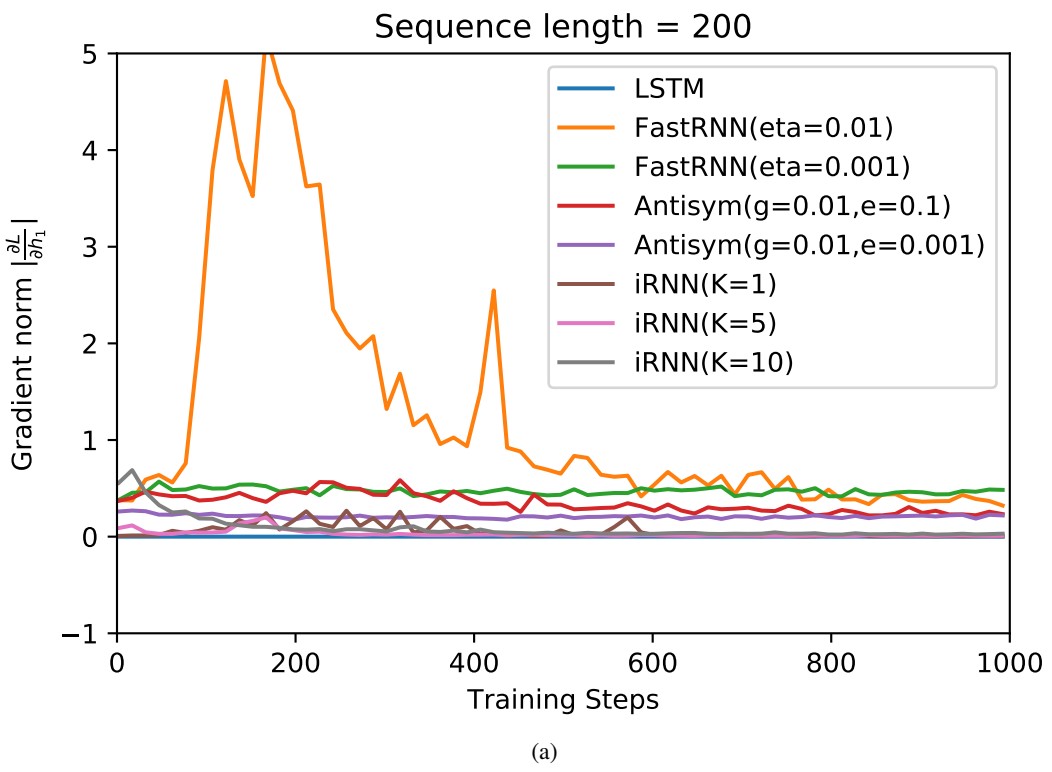

(a)

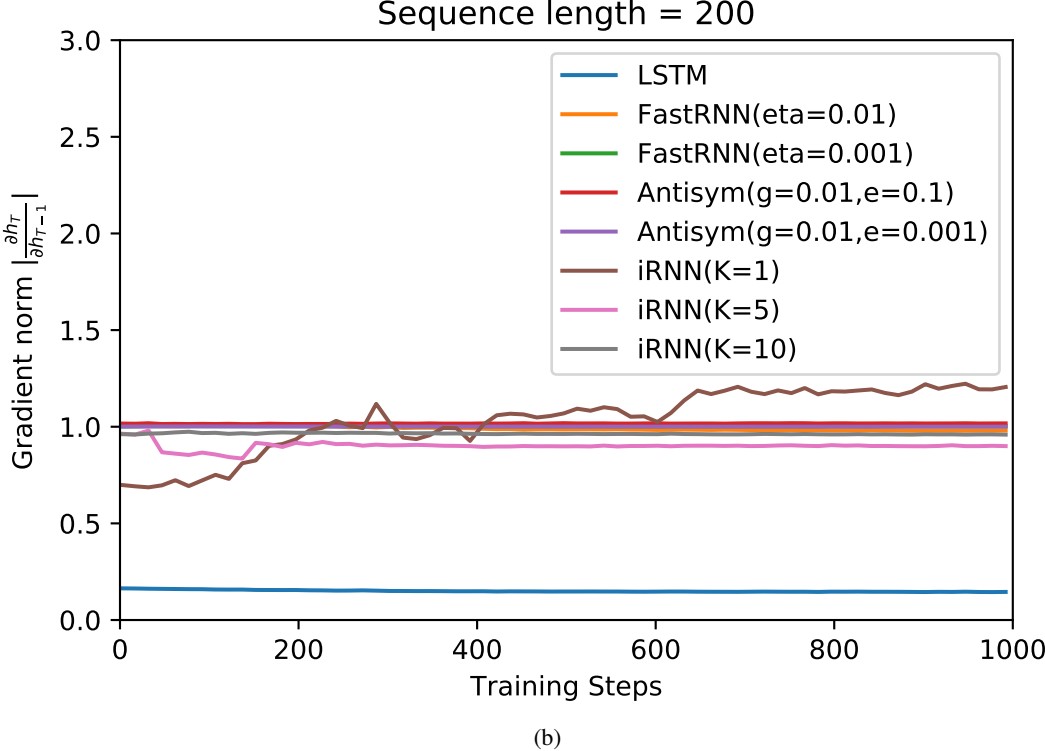

(b)

Figure 9: Exploratory experiments for the Add task : (a) Gradient norms w.r.t. loss $\|\frac{\partial L}{\partial h_1}\|$, (b) Gradient norms $\|\frac{\partial h_T}{\partial h_{T-1}}\|$. This together with Figure 3 shows that the gradients are identity everywhere for $K = 10$

**Theorem 3** (Local Convergence with Linear Rate). *Assume that the function $F$ in Eq. 12 and the parameter $\eta_k^{(i)}$ in Eq. 13 satisfy*

$$[\eta_k^{(i)}]^2 \|\nabla F(g_i)F(g_i)\|^2 + 2\eta_k^{(i)} F(g_i)^\top \nabla F(g_i)F(g_i) < 0, \forall i, \forall k. \qquad (18)$$

*Then there exists $\epsilon > 0$ such that if $\|g_0 - h_{eq}\| \leq \epsilon$ where $h_{eq}$ denotes the fixed point, the sequence $\{g_i\}$ generated by the Euler method converges to the equilibrium solution in $\mathcal{M}_{eq}(h_{k-1}, x_k)$ locally with linear rate.*

*Proof.* By substituting Eq. 16 into Eq. 17, to prove local convergence we need to guarantee

$$\|F(g_i) + \eta_k^{(i)} \nabla F(g_i)F(g_i)\| < \|F(g_i)\|. \qquad (19)$$

By taking the square of both sides in Eq. 19, we can show that Eq. 19 is equivalent to Eq. 18. We then complete our proof.

$$\square$$

**Corollary 2.** *Assume that $\|\mathbf{I} + \eta_k^{(i)} \nabla F(g_i)\| < 1, \forall i, \forall k$ holds. Then the forward propagation using Eq. 13 is stable and our sequence $\{g_i\}$ converges locally with linear rate.*

*Proof.* By substituting Eq. 16 into Eq. 17 and based on the assumption in the corollary, we have

$$\frac{\|r_i\|}{\|F(g_i)\|} = \frac{\|F(g_i) + \eta_k^{(i)} \nabla F(g_i)F(g_i)\|}{\|F(g_i)\|}$$

$$\leq \frac{\|\mathbf{I} + \eta_k^{(i)} \nabla F(g_i)\|\|F(g_i)\|}{\|F(g_i)\|} < 1. \qquad (20)$$

Further based on Prop. 2 in Chang et al. (2019) and Thm. 2, we then complete our proof. $\square$

Table 9: Results for Activity Recoginition Datasets.

| Data set | Algorithm | Accuracy (%) | Model Size (KB) | Train Time (hr) | Test Time (ms) | #Params |
|---|---|---|---|---|---|---|
| HAR-2 | FastRNN | 94.50 | 29 | 0.063 | **0.01** | 7.5k |
| | FastGRNN-LSQ | 95.38 | 29 | 0.081 | 0.03 | 7.5k |
| | FastGRNN | 95.59 | 3 | 0.10 | | |
| | RNN | 91.31 | 29 | 0.114 | **0.01** | 7.5k |
| | SpectralRNN | 95.48 | 525 | 0.730 | 0.04 | 134k |
| | EURNN | 93.11 | **12** | 0.740 | | |
| | LSTM | 93.65 | 74 | 0.183 | 0.04 | 16k |
| | GRU | 93.62 | 71 | 0.130 | 0.02 | 16k |
| | Antisymmetric | 93.15 | 29 | 0.087 | **0.01** | 7.5k |
| | UGRNN | 94.53 | 37 | 0.120 | | |
| | *iRNN* (K=1) | 95.32 | **17** | 0.061 | **0.01** | **4k** |
| | *iRNN* (K=3) | 95.52 | **17** | 0.081 | 0.02 | **4k** |
| | *iRNN* (K=5) | **96.30** | 18 | **0.018** | 0.03 | **4k** |
| DSA-19 | FastRNN | 84.14 | 97 | 0.032 | **0.01** | 17.5k |
| | FastGRNN-LSQ | 85.00 | 208 | 0.036 | 0.03 | 35k |
| | FastGRNN | 83.73 | 3.25 | 2.10m | | |
| | RNN | 71.68 | 20 | 0.019 | **0.01** | **3.5k** |
| | SpectralRNN | 80.37 | 50 | 0.038 | 0.02 | 8.8k |
| | LSTM | 84.84 | 526 | 0.043 | 0.06 | 92k |
| | GRU | 84.84 | 270 | 0.039 | 0.03 | 47k |
| | Antisymmetric | 85.37 | 32 | 0.031 | **0.01** | 8.3k |
| | UGRNN | 84.74 | 399 | 0.039 | | |
| | *iRNN* (K=1) | **88.11** | **19** | 0.015 | **0.01** | **3.5k** |
| | *iRNN* (K=3) | 85.20 | **19** | 0.020 | 0.02 | **3.5k** |
| | *iRNN* (K=5) | 87.37 | 20 | **0.005** | 0.03 | **3.5k** |
| Google-12 | FastRNN | 92.21 | 56 | 0.61 | **0.01** | 12k |
| | FastGRNN-LSQ | 93.18 | 57 | 0.63 | 0.03 | 12k |
| | FastGRNN | 92.10 | 5.5 | 0.75 | | |
| | RNN | 73.25 | 56 | 1.11 | **0.01** | 12k |
| | SpectralRNN | 91.59 | 228 | 19.0 | 0.05 | 49k |
| | EURNN | 76.79 | 210 | 120.00 | | |
| | LSTM | 92.30 | 212 | 1.36 | 0.05 | 45k |
| | GRU | 93.15 | 248 | 1.23 | 0.05 | 53k |
| | Antisymmetric | 89.91 | 57 | 0.71 | **0.01** | 12k |
| | UGRNN | 92.63 | 75 | 0.78 | | |
| | *iRNN* (K=1) | 93.93 | **36** | 0.20 | **0.01** | **8.1k** |
| | *iRNN* (K=3) | 94.16 | 37 | 0.33 | 0.03 | **8.1k** |
| | *iRNN* (K=5) | **94.71** | 38 | **0.17** | 0.05 | **8.1k** |
| Google-30 | FastRNN | 91.60 | 96 | 1.30 | **0.01** | 18k |
| | FastGRNN-LSQ | 92.03 | 45 | 1.41 | **0.01** | **8.5k** |
| | FastGRNN | 90.78 | 6.25 | 1.77 | | |
| | RNN | 80.05 | 63 | 2.13 | **0.01** | 12k |
| | SpectralRNN | 88.73 | 128 | 11.0 | 0.03 | 24k |
| | EURNN | 56.35 | 135 | 19.00 | | |
| | LSTM | 90.31 | 219 | 2.63 | 0.05 | 41k |
| | GRU | 91.41 | 257 | 2.70 | 0.05 | 48.5k |
| | Antisymmetric | 90.91 | 64 | 0.54 | **0.01** | 12k |
| | UGRNN | 90.54 | 260 | 2.11 | | |
| | *iRNN* (K=1) | 93.77 | **44** | 0.44 | **0.01** | **8.5k** |
| | *iRNN* (K=3) | 91.30 | **44** | 0.44 | 0.03 | **8.5k** |
| | *iRNN* (K=5) | **94.23** | 45 | **0.44** | 0.05 | **8.5k** |

| Algorithm | Test Perplexity | Model Size (KB) | Train Time (min) | Test Time (ms) | #Params |
|---|---|---|---|---|---|
| FastRNN | 127.76 | 513 | 11.20 | 1.2 | 52.5k |
| FastGRNN-LSQ | 115.92 | 513 | 12.53 | 1.5 | 52.5k |
| FastGRNN | 116.11 | 39 | 13.75 | | |
| RNN | 144.71 | **129** | 9.11 | **0.3** | **13.2k** |
| SpectralRNN | 130.20 | 242 | - | 0.6 | 24.8k |
| LSTM | 117.41 | 2052 | 13.52 | 4.8 | 210k |
| UGRNN | 119.71 | 256 | 11.12 | 0.6 | 26.3k |
| *iRNN* (**K=1**) | **115.71** | 288 | **7.11** | 0.6 | 29.5k |

Table 10: PTB Language Modeling: 1 Layer. To be consistent with our other experiments we used a low-dim **U**; For this size our results did not significantly improve with $K$. This is the dataset of Kusupati et al. (2018) which uses sequence length 300 as opposed to 30 in the conventional PTB.

| Data set | Algorithm | Accuracy (%) | Model Size (KB) | Train Time (hr) | Activation | #Params |
|---|---|---|---|---|---|---|
| HAR-2 | *iRNN* (K=1) | 95.32 | **17** | 0.061 | ReLU | **4k** |
| | *iRNN* (K=3) | 95.52 | **17** | 0.081 | ReLU | **4k** |
| | *iRNN* (**K=5**) | **96.30** | 18 | **0.018** | ReLU | **4k** |
| | *iRNN* (K=1) | 92.16 | **17** | 0.065 | Sigmoid | **4k** |
| | *iRNN* (K=3) | 93.35 | **17** | 0.078 | Sigmoid | **4k** |
| | *iRNN* (**K=5**) | 95.30 | 18 | 0.020 | Sigmoid | **4k** |

Table 11: HAR-2 dataset (Sigmoid, ReLU activations): $K$ denotes pre-defined recursions embedded in graph to reach equillibrium.

