# OpenReview forum: "RNNs Incrementally Evolving on an Equilibrium Manifold: A Panacea for Vanishing and Exploding Gradients?"
_ICLR.cc/2020/Conference — Accept (Poster)_

### Official Review · AnonReviewer2 · 2019-10-22
**Official Blind Review #2**

**Rating:** 6

**Review:**

The authors present a novel work to address the problem of signal propagation in the recurrent neural networks. The idea is to build a attractor system for the signal transition from state h_{k-1} to h_k. If the attractor system converges to a equilibrium, then the hidden to hidden gradient is an identity matrix. This idea is elegant. The authors verify the performance of Increment RNN on long-term-dependency tasks and non-long-term-dependency tasks.

The work successfully constructs a negative identity hidden to hidden gradient matrix but I still have a few concerns about the theory and the experiments. If the authors can address my concerns in the rebuttal, I am willing to increase my score.

Theoretical concerns:
Even in the limit sense, the inner ordinary differential equation of g will converge to the equilibrium, it may not converge in the finite steps. Thus, the state-to-state gradient may be slightly away from the identity. And we know that (0.99)^T goes to infinity when T goes to infinity. In practice, the long-term-gradient problem may still exist in the incremental RNN.

Experimental concerns:
The theorem requires the norm of U to be bounded. But I cannot see how the authors bound the norm of U in the experiment section.

Clarity of writing:
There are a bunch of typos in the papers, especially the ones in the proof of Theorem 1. The proof of theorem 1 can be polished. The author swap the use of phi and psi several times in the proof. And gradient calculation on equation (5) should be expanded into more details. The authors also missed one U in the equation in the second last line of the proof.

Overall, I think the paper is an interesting contribution to the community.


**Experience Assessment:**

I have read many papers in this area.

**Review Assessment: Checking Correctness Of Derivations And Theory:**

I carefully checked the derivations and theory.

**Review Assessment: Checking Correctness Of Experiments:**

I carefully checked the experiments.

**Review Assessment: Thoroughness In Paper Reading:**

I read the paper thoroughly.

---

> ### Author Response · Authors · 2019-11-08
> **Uploaded paper addresses finite K and gradient issue, theorem 1 typos, and experimental issues arising from constraint on $U$ required in theorem. A number of remarks and comments have been introduced to address reviewer comments.**
>
> We thank the reviewer for constructive feedback. We updated the paper with suggested changes.
>
> 1. Gradient is not Identity for finite $K$ and $0.99^T$ will still result in vanishing gradient.
>
> This is an important observation and is central to our method. The key idea is to choose $K$ as a function of $T$. While larger $T$ degrades gradients, it is more than compensated by larger values of $K$. Fundamentally, this is because we achieve equilibrium at a geometric rate.
>
> To ensure that end-to-end partials are no worse than $1\pm \epsilon$, we need $K = O(\log(T/\epsilon)$.  As such $K=\log(T/\epsilon)$ is typically small and manageable. That this choice works follows from geometric rate of convergence of the partials (under additional second-order conditions on $\phi$), namely, we get $\|{\partial g_K \over \partial h_m} - I\| \leq (1-\eta (\alpha - \|U\|))^K$.
>
> It implies that over T-fold time steps, we have $(1 \pm (1-\eta (\alpha - \|U\|))^K)^T$ as the end-to-end gradient. We can bound this as no worse than $1 \pm T \exp(-K \log(1-\eta(\alpha-\|U\|))$. Direct computation shows that we need to set the value of $K \approx \log(T/\epsilon)$.
>
> 2. Handling $\|U\| \leq \alpha$ in experiments.
>
> This is an important issue. In experiments, we set $\alpha=1$, and additionally do not enforce $\|U\|< \alpha$ constraint. Instead, we initialize $U$ as a Gaussian matrix with IID mean zero, small variance components. As such, the matrix norm is significantly smaller than 1 to start with. Evidently, the resulting learnt $U$ matrix does not violate this condition, and as such this choice does not appear to degrade our performance. Nevertheless, we agree that imposing this constraint may lead to additional overhead.
>
> 3. Theorem 1 Edits
>
> We introduced an informal gradient computation in Sec. 3. As such the specific gradient computation is worked out there. We also corrected other typos that the reviewer identified.
>
> We appreciate the feedback.

---

> > ### Comment · AnonReviewer2 · 2019-11-15
> > **The authors' feedback does not fully address my questions; but I still vote for acceptance.**
> >
> > Thanks the authors for the detailed feedback. The feedback is honest but it does not address my questions.
> >
> > 1. I am not sure $1 \pm T \exp(-K \log(1-\eta(\alpha-\|U\|))$ infers that $K \approx \log(T/\epsilon)$. $\log(1-\eta(\alpha-\|U\|))$ can be zero if $\alpha-\|U\|=0$. Also if K is large, why not just parametrize the model with a unitary matrix like in URNN?
> >
> > Arjovsky, Martin, Amar Shah, and Yoshua Bengio. "Unitary evolution recurrent neural networks." International Conference on Machine Learning. 2016.
> >
> > 2. [Evidently, the resulting learnt  matrix does not violate this condition, and as such this choice does not appear to degrade our performance. ]
> > This sentence is loose. Maybe something else in the model helps the performance.
> >
> > In general, besides the problems I mentioned, I still think it is an interesting paper but I will not increase my score.

---

> > > ### Author Response · Authors · 2019-11-15
> > > **Response regarding our work and Unitary RNN and other comments**
> > >
> > > We again thank the reviewer for the detailed feedback. We feel that we did not emphasize a central point in our response sufficiently. Perhaps identity gradient is over-emphasized, but in reality what we seek is that over the length T, we can circumvent vanishing/exploding gradient. This means that we neither want gradient to go to zero or approach infinity. Identity gradient is nice but not necessary in this context. Perhaps our response below might clarify the issue a bit.
> > >
> > > 1. What is implication that our end-to-end gradient is in the interval  $[1- T \exp(-K \log(1-\eta(\alpha - \|U\|)), 1+ T \exp(-K \log(1-\eta(\alpha - \|U\|))] \triangleq [1\pm T \exp(-K \log(1-\eta(\alpha - \|U\|))]$?
> > >
> > > Perhaps this got missed in our response. We will show that exploding/vanishing gradient can be circumvented if $K$ were chosen as $\log(T)$.
> > >
> > > This means that our objective is to ensure that the product of gradients are within $1\pm \epsilon$, where $\epsilon$ is some small number but need not be too small. For instance, $\epsilon = 1/2$ is also ok. This is possible by choosing $K \approx \log(T/\epsilon)$.
> > >
> > >
> > > Remark: We do not want $\|U\| - \alpha=0$ as we argue below, contrary to reviewer's comment. Indeed, from the expression above $\|U\|=\alpha$ is problematic. Perhaps, reviewer is relating this to unitary RNN, where $\|U\|=1$ is desired, but our method is unrelated to unitary rnn. In reality we want $\|U\| \ll \alpha$.
> > >
> > > Consider,
> > >
> > >  \begin{align}
> > > 	\frac{\partial h_r}{\partial h_s} = \prod_{r\geq m>s}\frac{\partial h_m}{\partial h_{m-1}} \in [1\pm T \exp(-K \log(1-\eta(\alpha - \|U\|))] \in [1 \pm \epsilon]
> > > 	\end{align}
> > >
> > > To do this, we let $T \exp(-K \log(1-\eta(\alpha - \|U\|)) \leq \epsilon$. This implies that
> > > $ \log(T/\epsilon) \leq K(- \log(1-\eta(\alpha - \|U\|))$ and so,
> > > $$
> > > K \geq \frac{1}{-\log(1-\eta(\alpha - \|U\|))} \log(T/\epsilon)
> > > $$
> > > This suggests that we do not want $\|U\|=\alpha$ but somewhat smaller $\|U\|$. The point is that we can choose parameter $\eta, \alpha, \|U\|, K$ so that the end-to-end gradient is neither vanishing or exploding. To do so, we only need $K$ as a logarithmic function of $T$. What we pointed out in our earlier response was that this was entirely manageable, if all we are interested in is to circumvent vanishing/exploding gradients.
> > >
> > >
> > > 2. Resulting learnt matrix does not violate condition is too loose.
> > >
> > > For a small value of $\eta$, and a bounded number of epochs, while  $U$ trajectory during training has a strong directional component, we may not necessarily move very far in magnitude from the initialized $\|U\|$. Consequently, the constraint as such did not arise.
> > >
> > >
> > > 3. Difference between Unitary RNN and this paper.
> > >
> > > Fundamentally following are the differences. We have pointed this out in the paper.
> > >
> > > (a) We avoid both exploding and vanishing gradients, albeit with K growing logarithmically in T. Unitary RNN can fix the exploding gradient issue but vanishing gradient is still a problem.
> > >
> > > (b) Our method involves an inner recursion. Since $K=O(\log(T)$, computational scaling is somewhat close to conventional RNN. In other words, inference time scales as (RNN_Time)$\log(T)$. Training is interesting because we actually do better than RNN. This is because, while we lose $\log(T)$ factor we gain significantly due to circumventing vanishing/exploding issue of RNNs.
> > >
> > > For unitary RNN, unitarity has to maintained at each step, a computationally cumbersome step, as many others have noted.
> > >
> > > (c) The transition, $h_t = \phi(U h_{t-1} + Wx_t)$ with $U$ unitary does not imply partial gradients are unity as argued in the unitary RNN (Eq. 3). In contrast, in theory we can ensure identity gradients. Nevertheless, practically, we can circumvent vanishing/exploding gradient over T steps with logarithmic K.

---

### Official Review · AnonReviewer1 · 2019-10-23
**Official Blind Review #1**

**Rating:** 6

**Review:**

In this paper, the authors propose the incremental RNN (iRNN), which is inspired by the continuous-time RNN (CTRNN). Theoretically, the equilibrium point of iRNN exists and is unique. Furthermore, the norm of the Jacobian between two hidden states is always one, provided that the Euler iterations converge. The authors proved this property as well as the exponential convergence rate of the Euler iteration. These properties avoid the vanishing/exploding gradient problem typical in RNN with long sequences in theory. Empirically, the proposed method is compared with multiple RNN architecture on various tasks.

The paper is clearly written and well organized. It starts with the existence and uniqueness of a fixed point; then provides the converge rate, and finally the main theorem of identity gradient. The main theorem provides a good theoretically guaranteed solution. The authors have done extensive experiments on several datasets compared against various popular and recently developed RNN-based models. The authors also provided some empirical experiments on the converge rate and the non-singularity of grad \psi.

The motivation of the paper could be improved. For example, despite the appealing theoretical properties. I did not fully understand the motivation in Equation 3. It would be great if the authors could further elaborate on it.

Typos:
Page 4 at “Upon computation, we see that”. The gradient on the left-hand side should be grad \psi?
Page 4 at “at the equilibrium points we have”, the gradient of \psi should be the gradient of \phi?


**Experience Assessment:**

I have published one or two papers in this area.

**Review Assessment: Checking Correctness Of Derivations And Theory:**

I assessed the sensibility of the derivations and theory.

**Review Assessment: Checking Correctness Of Experiments:**

I assessed the sensibility of the experiments.

**Review Assessment: Thoroughness In Paper Reading:**

I read the paper at least twice and used my best judgement in assessing the paper.

---

> ### Author Response · Authors · 2019-11-08
> **Uploaded paper addresses comments about motivation.**
>
> We thank the reviewer for his/her comments. We updated the paper with suggested changes and corrected the typos.
>
> 1. Motivation
>
> We added the following example in the introduction. We hope this is what the reviewer is looking for. We will appreciate additional feedback.
>
> Consider a discrete sequence, $x_m$ and the corresponding hidden-states, $h_m$. These $h_m$'s are for CTRNN in equilibrium (namely $\beta \gg 1$). This means that $$-\alpha(g(\beta \tau) + h_{m-1}) + \phi(U (g(\beta \tau) + h_{m-1}) + W x_m + b)=0$$
> Denoting the equilibrium solution, as $\mu(x_m,h_{m-1})$, we have, $-\alpha(\mu(x_m,h_{m-1})) + \phi(U (\mu(x_m,h_{m-1})) + W x_m + b)=0.$
>
> We see that state transitions are then marginal changes from previous states, namely,
> $$h_m = \mu(x_m,h_{m-1}) - h_{m-1}$$
> Now for a fixed input $x_m$, as to which equilibrium is reached depends on $h_{m-1}$, but there are nevertheless finitely many equilibrium points. So encoding marginal changes as states leads to "``identity" gradient.
>
> Intuitively,  $h_m$ refers to the marginal change in the equilibrium solution from the previous state and depends on new information in the input $x_m$. It is responsive to sudden changes in the input. Note that $h_m$ depends on data $x_1,x_2,\ldots, x_m$, while $h_{m-1}$ on $x_1,x_2,\ldots, x_{m-1}$, and so the marginal change could be viewed as ``new information'' that needs to be accounted for.
>
> 2. Typos
>
> We corrected the indicated typos.

---

### Official Review · AnonReviewer3 · 2019-10-23
**Official Blind Review #3**

**Rating:** 8

**Review:**

Update: the authors have addressed my issues below and I have raised my score to 8.

***

This paper on recurrent neural networks goes back to Rosenblatt's continuous time dynamics model and uses a discretised version of that equation (equation (1) in the paper) to build an incremental version of the RNN, called incremental RNN, where the transition from hidden state h_{k-1} at step k-1 to next step's h_k is done using small incremental updates until the system achieves equilibrium. It claims that it manages to solve the vanishing gradient problem by keeping all gradients \frac{\partial h_k}{\partial h_{k-1}} equal to minus identity matrix. The algorithm is then extensively evaluated on a large number of tasks and compared with plain RNNs, LSTMs, and two recently published papers on antisymmetric RNN and FastRNN.

I need to admit that after reading the paper twice, I am not sure I understand how the method works exactly (how does inserting intermediary steps and variables g_0, g_1, ... g_T enable the system to reach equilibrium: is there an iterative evaluation until convergence?) and more worringly, how the single-step SiRNN differs from a normal RNN with an extra residual connection?

According to equation (5), for T=1 and g_0=0, we have:
g_T = g_1 = \eta_k^1 ( \phi (U (h_{k-1}) + W x_k + b) - \alpha h_{k-1} ).
If the gradients are vanishing in the normal RNN, why would they not vanish here for T=1?

Propositions 1 and 2 are for the case K=\infinity, and I could not understand the proof of theorem 1 that shows why \frac{\partial h_k}{\partial h_{k-1}} = - I. This seems to be the major contribution of the paper and should be given prominence.

What is missing is clear explanation, like (Bengio et al, 1994), of the identity gradient and of how the algorithm works. These questions could be solved by including code or pseudo-code explaining how to actually implement incremental RNNs.

There are also several important papers recently published that have approached the problems of continous-time dynamics and relaxation of hidden state to equilibria.
* The paper does not mention at all Neural ODEs [2] [3] where the state flows in a continuously differentiable way thanks to the continuous-time residual network ODE formulation. Moreover, isn't the idea of inserting a relaxation to equilibrium using ODEs already implemented in the ODE-RNNs [3]?
* How do incremental updates related to Adaptive Computation Time [4]?

For this reason, I am currently tending to reject the paper, but am open to change my score upon clarifications and links to other similar work.

Additional remarks:
The first paragraph of the paper explains the Elman RNN, not RNNs in general.
Please cite [1] alongside Bengio et al (1994) for the problem of the vanishing gradient.
Define alpha in equation (1)
Notation k and K is very confusing
Blue vs. green on Figure 2 is hard to read, and where is the new initialisation?
Why do you add h_k^K to h_{k-1} in equation (8)? I thought it was g_k^K?
Keep the same colours for all experiments in figure 4.

[1] Hochreiter (1991) "Untersuchungen zu dynamischen neuronalen Netzen"
[2] Chen, Rubanova, Bettencourt & Duvenaud (2018) "Neural Ordinary Differential Equations"
[3] Rubanova, Chen & Duvenaud (2019) "Latent odes for irregularly-sampled time series"
[4] Graves (2016) "Adaptive computation time for recurrent neural networks"


**Experience Assessment:**

I have published one or two papers in this area.

**Review Assessment: Checking Correctness Of Derivations And Theory:**

I assessed the sensibility of the derivations and theory.

**Review Assessment: Checking Correctness Of Experiments:**

I assessed the sensibility of the experiments.

**Review Assessment: Thoroughness In Paper Reading:**

I read the paper at least twice and used my best judgement in assessing the paper.

---

> ### Author Response · Authors · 2019-11-08
> **Reviewer comments have been addressed in uploaded paper. Pseudo code has been uploaded; schematics of algorithm is introduced; proof of theorem 1 clarified; extensive discussion on related work is presented; and a number of remarks introduced to clarify some of nuances of the work.**
>
> We thank the reviewer for constructive feedback. We expanded Sec 3 (Methods) to expose some of the issues that were not transparent.
>
> **Additional Comments: k and K was confusing. So we now have k for recursion and m for time. $\alpha$ is defined. Figs are updated. Eq. 8 (now 9) is updated.
>
> 1. Algorithm, Pseudo Code, Implementation, figure.
>
> We included psuedo-code in appendix and introduced figure 1 in Sec. 3. A new comment (root-finding and transitions) is introduced to describe the algorithm. The algorithm has two indices $k$ and $m$, and serve different purposes. . The index $m \in [T]$ refers to the time index, and indexes input, $x_m$ and hidden state $h_m$ over time horizon $T$. The index $k \in [K]$ is used for the fixed-point recursion for converging to the equilibrium point for each time $m$. $g_k$ is the only variable that is indexed with $k$ here and we iterate over $g_k$'s so that, for $k=K$, we have
> $$\phi(U(g_{K}+h_{m-1}) + W x_m + b) - \alpha(g_{K}+ h_{m-1}) \approx 0$$
>
> The recursion (Eq.~5) at time $m$ runs for $K$ rounds, terminates, and recursion is reset for the new input, $x_{m+1}$. Eq.~5 is a standard root-finding recursion. $g_{k-1}$ is the previous solution, which is modified by a correction term: the error, $\phi(U(g_{k-1}+h_{m-1}) + W x_m + b) - \alpha(g_{k-1}+ h_{m-1})$. If the sequence converges, the resulting solution is the equilibrium point. Proposition~2 guarantees a geometric rate of convergence.
>
> 2. According to equation (5), for T=1 and g_0=0, we have: ...
> If the gradients are vanishing in the normal RNN, why would they not vanish here for T=1?
>
> As we pointed out in response above, there are two indices, $k$ and $m$. The index $k$ runs for $K$ rounds, and guarantees that at each m, $g_k$ is iterated, while fixing $m$. At each $m$, ${\partial h_m \over \partial h_{m-1}}$ is identity if K approaches infinity.
>
> Perhaps the reviewer is referring to the case $K=1$ and SiRNN. In this case we do not claim that gradient is identity. All we say is that if the input is slowly varying, we expect to track iRNN and thus can expect to have close to identity gradients.
>
> 3. Worringly SiRNN looks like Residual connections.
>
> We introduced a new remark to clarify this point. As such, our architecture is a special case of skip/residual connections. Nevertheless, unlike skip connections, our connections are {\it structured}, and the dynamics driven by the error term (see 1 above) ensures that the hidden state is associated with equilibrium and leads to identity gradient. No such guarantees are possible with unstructured skip connections. Note that for slowly varying inputs, after a certain transition-time period, we should expect SiRNN to be close to equilibrium as well. Without this unstructured residual connections can learn patterns that can be dramatically different (see Fig.~2).
>
> 4. I could not understand the proof of theorem 1.
>
> This is perhaps because the algorithm was not clear. We introduced a comment that goes through the derivation in Sec. 3. Basically, the point is that when the system is in equilibrium, it must satisfy $\phi(U(h_{m-1} + g_K) + Wx_m + b) - \alpha (h_{m-1} + g_K) = 0$. This is a surface that relates $h_{m-1}$ and $g_K$. Note that $h_m = g_K$. In this comment, we explicitly derive the gradient for one-transition and show ${\partial h_m \over \partial h_{m-1}} = -I$. So now Theorem~1's proof is clearer. Our Theorem~1 statement is application of chain-rule and pointing out that product of identity is identity.
>
> 5. Related Works.
>
> Thanks for pointing them out. We now discuss them in our related work section. [2] and [3]'s focus is somewhat different.
>
> In the context of time-series prediction, their goal is to account for irregularly sampled inputs. To do this they parameterize the derivative of the hidden-state in terms of an {\it autonomous} differential equation, i.e., without any input. The benefit of the ODE is that the system evolves in continuous time oblivious to the gap between any two input samples. Then, when the next input arrives the internal state can be updated.
>
> Our method is different. We explicitly include inputs into our ODE and reach an equilibrium solution in response to that input. Then, we encode marginal changes between the equilibrium solution and previous state to account for input variations. We prove that this setting results in identity gradient. While obviating vanishing/exploding gradient is not of concern to [2] and [3], no such guarantees exist in their context.
>
> [4] indeed resembles our setting. However, again their work is complementary. They propose to vary $K$ for different times $m$ as a way to attend to important input transitions. However, the transition functions they use can be gated units, while we primarily utilize conventional functions. Furthermore, while this is not their concern, as such equilibrium may not even exist and identity gradients are not guaranteed, in their setup.

---

> > ### Comment · AnonReviewer3 · 2019-11-15
> > **Official Blind Review #3 update**
> >
> > Thank you for all the clarifications and links to Neural ODEs and ACT. The paper is much improved and I have raised my score accordingly.

---

### Decision · Program_Chairs · 2019-12-19

**Decision:**

Accept (Poster)

**Comment:**

In this paper, the authors propose the incremental RNN, a novel recurrent neural network architecture that resolves the exploding/vanishing gradient problem. While the reviewers initially had various concerns, the paper has been substantially improved during the discussion period and all questions by the reviewers have been resolved. The main idea of the paper is elegant, the theoretical results interesting, and the empirical evaluation extensive. The reviewers and the AC recommend acceptance of this paper to ICLR-2020.